

# Monitoring snow depth change across a range of landscapes with ephemeral snow packs using Structure from Motion applied to lightweight unmanned aerial vehicle videos

Richard Fernandes[1], Christian Prevost[1], Francis Canisius[1], Sylvain G. Leblanc[1], Matt Maloley[1], Sarah
Oakes[1], Kiyomi Holman[2], Anders Knudby[2]

[1]Canada Centre for Remote Sensing, Natural Resources Canada, Ottawa, K1A 0Y7, Canada
[2]Department of Geography, Environment and Geomatics, University of Ottawa, Ottawa, K1N 6Y5, Canada

*Correspondence to*: Richard Fernandes (richard.fernandes@canada.ca)

**Abstract.** Snow depth (SD) can vary by more than an order of magnitude over length scales of metres due to topography, vegetation and microclimate. Differencing of digital surface models derived from Structure from Motion (SfM) processing of airborne imagery has been used to produce SD maps with between ~2 cm to ~15 cm horizontal resolution and accuracies on the order of +/-10 cm over both relatively flat surfaces with little or no vegetation and over alpine regions. Studies indicate that accuracy is lower in the presence of vegetation above or below the snowpack and in rough topography; suggesting that some biases may be temporally persistent. Moreover, flight and image parameters vary across studies but they are typically not related *a priori* to an expected uncertainty in SD. This study tests two hypotheses: i) that SD change can be more accurately estimated when differencing snow covered elevation surfaces rather than the absolute snow depth based on differencing a snow covered and snow free surface and ii) the vertical accuracy of SfM processing of imagery acquired by commercial light weight unmanned aerial vehicle (UAV) systems can be adequately modelled using conventional photogrammetric theory. Moreover, these hypotheses are tested over areas with ephemeral snow pack conditions and across a range of micro-topography and vegetation cover. Weekly SD maps with <3 cm horizontal resolution are derived for a period spanning peak snowpack to snow free condition for five sites with differing micro-topography and vegetation cover. Across the sites, the root mean square difference (RMSD) over the observation period, in-comparison to the average of in-situ measurements along ~50 m transects, ranged from 1.58 cm to 10.56 cm for SD and from 2.54 cm to 8.68 cm for weekly SD change. RMSD was not related to micro-topography as quantified by the snow free surface roughness. Biases in SD due to vegetation in the snow covered or snow free image contributed to over 85% of the observed difference at sites where the RMSD of SD exceed 5 cm. RMSD of weekly SD change was dominated by outliers corresponding to rapid in-situ melt or onset and low point cloud density. In contrast to the RMSD, the median absolute difference of SD change ranged from 0.65 cm to 2.71 cm. Validation results agree with photogrammetric theory that predicts uncertainty is proportional to UAV altitude and linearly related to horizontal uncertainty. These results indicate that while the accuracy of UAV based estimates of snow depth is similar to other studies with different snow pack and terrain conditions, the accuracy of UAV based estimates of weekly snow depth change was, excepting conditions with low point cloud densities, substantially better and is comparable to in-situ methods.





# 1 Introduction

The temporal and spatial pattern of snow depth (SD) is of importance to hydrological, ecological and climate studies (GCOS, 2016). Together with representative estimates of snow density, time series of SD are indicative of changes in snow water
equivalent that in turn are of importance to streamflow forecasting and management of hydroelectric resources (Clyde, 1939; Barnett et al., 2005; DeWall and Rango, 2008). In many ecosystems, SD is an important determinant of winter habitat in terms of range and access to forage (Bokhorst et al., 2016). Snow depth also exerts an influence on local climate through insulation of permafrost and ice and global climate through its role in snow albedo feedbacks (IPCC, 2013; IPCC, 2014; Bokhorst et al., 2016).

The World Meteorological Organization (WMO) coordinates a global synoptic in-situ SD monitoring network (https://globalcryospherewatch.org/projects/snowreporting.html) providing sub-daily measurements using rigorous standards (https://library.wmo.int/pmb_ged/wmo_8_en-2012.pdf ). This network is supplemented by public and private sector in-situ SD monitoring networks using protocols and instruments similar to the WMO synoptic network (e.g. Worley et al., 2015;
Reges et al., 2016). Automated in-situ instruments within these networks include acoustic and laser ranging devices that are typically fixed in location and that have spatial sampling footprints from 1 $m^2$ to 10 $m^2$ (e.g. Ryan et al., 2008; de Haij, 2011) and global positioning system (GPS) instruments that can estimate the mean snow depth over a footprint of ~$10^4$ $m^2$ (Larson et al., 2014). Manual measurements within in-situ networks are performed using rulers. The WMO protocol for ruler based sampling requires between one and six stakes, each having a spatial footprint of ~$10^{-2}$ $m^2$, spaced at intervals between 5 m and
10 m (WMO, 1970). The WMO also specifies supplementary snow courses for occasional measurement with between 50 and 100 locations along a transect where an observer inserts a ruler. Other networks use different ruler sampling protocols with plots or transects covering between 10 $m^2$ and 1000 $m^2$ although the total sampled footprint is actually under 10 $m^2$ (US Department of Commerce, 1997; Ryan et al., 2008; Meteorological Service of Canada, 2016).

While in-situ monitoring networks offer frequent temporal sampling, with the exception of GPS approaches, their spatial sampling can be imprecise and are often biased in terms of their representativeness of surrounding landscapes (Gelfan et al., 2004; Essery and Pomeroy, 2004; Neumann et al., 2010; Wrzesien et al., 2017). GPS survey may offer a solution for an average SD estimation over open terrain although measurement error is larger than manual methods (e.g. Larson et al. (2014) report bias and precision of -5.7 cm and 10.3 cm respectively when estimating SD of a snowpack typically under 1 m in depth).
Irrespective of measurement method, SD monitoring sites usually require road and/or power access; often leading to their co-location with low lying built up areas, airports or weather stations located along mountain tops (Brown et al., 2003). These locations can have vastly different microclimates and topographic conditions than less accessible areas nearby thus increasing the potential for biases in estimated SD.



One solution to address the limitation of sparse and potentially spatially biased in-situ SD monitoring is to estimate the spatiotemporal SD pattern by combining in-situ SD time series and maps of SD change ($\Delta SD$) derived from remote sensing methods (e.g. Liu et al., 2017). Two research questions must be addressed to determine if this solution is viable. First, is it

possible to provide non-destructive on-demand spatial survey of $\Delta SD$ with uncertainty comparable to that of estimates from in-situ instruments over the same spatial footprint? Second, how well can sparse temporal samples of SD patterns together with dense temporal sampling of $\Delta SD$ at a point(s) reconstruct the temporal evolution of $\Delta SD$ over a region? This paper addresses the first research question over relatively open areas of varying micro-topography and snowpack conditions as a precondition for addressing SD mapping over arbitrary surface conditions.

Remote *SD* mapping at a similar or better resolution of automated in-situ measurements (i.e. <1 m$^2$) can be performed using airborne survey with LIDAR (e.g. Deems et al., 2013) or photogrammetric imaging (e.g. Nolan et al., 2015). Here we consider photogrammetric imaging approaches due to both their potential cost effectiveness and the widespread availability of unmanned aerial vehicle (UAV) systems. Nolan et al. (2015) used Structure from Motion (SfM; Westoby et al., 2012)

processing of 15 cm ground sampling distance (GSD) digital images from a manned aircraft at an altitude of ~750m above ground level (a.g.l.) to map *SD* with an accuracy (precision) of +/-10 cm (8 cm at 1 standard deviation) in comparison to individual probe measurements over relatively flat surfaces. Similar results were subsequently reported using UAV systems, with GSD ranging from ~2cm to ~10cm and altitude from 60 m a.g.l. to 130 m a.g.l., over prairies (Harder et al., 2016), alpine shrub lands (Buhler et al., 2016; De Michele et al., 2016; Harder et al., 2016; Avanzi et al., 2017) and glaciers (Gindraux et

al., 2017). Even greater accuracy (1.5 cm to 3.8 cm) and precision (4.2 cm to 9.8 cm at 1 standard deviation) have been reported for $\Delta SD$ mapping over tundra (Cimoli et al., 2015) and alpine terrain (Vander Jagt et al., 2015) when using very low (10 m a.g.l. – 30 m a.g.l.) altitude acquisitions with GSD less than 4 cm.

While current studies provide increasing evidence of the potential for *SD* mapping over certain landscapes using multi-date

UAV imagery there are a number of issues that must be addressed if this approach is to be applicable for routine seasonal estimation of *SD* or $\Delta SD$ over natural landscapes. A pressing issue is the need to test the performance of this approach over a range of snowpack, vegetation and terrain conditions (de Michele et al., 2016). Studies indicate the presence of large (>10cm) errors under specific illumination, snowpack, vegetation or terrain conditions. The reduced contrast in imagery of homogenous snowpacks (due to fresh snow covering all vegetation) under overcast conditions results in reduced point cloud density (Nolan

et al., 2015; Buhler et al., 2017) and can lead to the failure of commercial SfM algorithms (Harder et al., 2016). While this issue may be partly addressed by using both visible and near-infrared imaging (Buhler et al., 2017) it may also be less of a factor when there is structure in the snowpack due to emergent vegetation and when GSD is sufficiently high to identify the intersection of snow and vegetation. Dense low vegetation compressed by the snowpack can result in SD underestimates due to a positive elevation bias in the snow free reference image (Nolan et al., 2015; Buhler et al., 2016; Cimoli et al., 2015; Di





Michele et al., 2016).   Vegetation above the snowpack can result in local overestimates of SD if they are incorrectly interpreted as the snowpack surface (Nolan et al., 2015; Harder et al., 2016).  Topographic shadowing can have the same impact as overcast conditions when estimating SD over homogenous snow packs (Buhler et al., 2017).  However, the shading from vegetation and micro-topography on $SD$ estimates has not been studied systematically in the sense of considering different terrain

roughness under the same snowpack and acquisition conditions.

A second issue that has yet to be addressed is the performance of UAV imaging approaches for estimating $\Delta SD$ between two dates with partial or complete snow cover. Current UAV imaging methods may have a practical lower limit of ~30cm $SD$ due to the combined errors in estimating the snow covered and snow free surface elevation (Harder et al., 2016).  However, in

many circumstances $\Delta SD$ may still have relevance (e.g. for temporal monitoring or for estimating $SD$ using a reference snow covered date where $SD$ is known using in-situ methods).  Errors due to factors such as vegetation and terrain may be spatially correlated so that estimates of $\Delta SD$ between short periods of time (e.g. weekly) may be substantially more accurate that estimates of SD itself.  There is a need to compare the relative accuracy and temporal precision of $SD$ and $\Delta SD$ estimates, especially for areas with ephemeral snow packs.

A third issue is the need to model the uncertainty of elevation estimates as a function of UAV mission parameters.  This is required both to guide mission parameters and to understand the potential limits of current technologies and prospects for improvements as UAV performance and camera systems improve.  Nasrullah (2016) demonstrated that photogrammetric theory can be used for this purpose when estimating the elevation of man-made targets using UAV imagery and SfM over

man-made targets. A similar modelling approach has yet to be tested over snow covered surfaces.

A fourth issue is the need to have robust low-cost equipment and software for data acquisition and processing (Nolan et al., 2015).  Light weight systems that require minimal flight certification are especially desirable considering that snow surveys may be episodic both in time and space.  Nasrullah (2016) found that using commercial SfM software (Pix4D Version 2.1.100)

with imagery from off-the-shelf UAV systems weighing less than 2kg and costing under $US 1000  (Phantom 2 Vision+) provided comparable performance to larger drones.  There is a need to evaluate similar systems for $\Delta SD$ mapping over a range of environmental and surface conditions.

The issues that remain to be addressed regarding UAV based mapping of $\Delta SD$ require multiple experimental treatments

including climate and snow conditions that cannot easily be controlled and land surface conditions that can be controlled.  Here we control the survey methodology by using a single low-cost light weight commercially available solution for UAV based mapping of three dimensional point clouds and select mission parameters that should maximize the accuracy of elevation estimation based on photogrammetric theory, even if the solution may not be optimal in the sense of logistical constraints of time or cost.  Secondly we select sites with a range of micro-topography and vegetation cover but limit vegetation cover to

<50% and only validate SD in openings. This strategy simplifies the approach used to extract surface locations within three dimensional point clouds leaving the issue of UAV based SD mapping under closed canopies for further study. Thirdly we locate the sites within regions of ephemeral snowpack since this should correspond to a worst case assessment of uncertainty, especially with respect to $\Delta SD$. Given these limitations, the initial broad research question regarding snow depth mapping is refined into two specific research questions addressed in this study:

What is the accuracy and precision of $SD$ and weekly $\Delta SD$ maps derived using small commercial UAV and commercial SfM technology as a function of varying micro-topography and snowpack condition over natural landscapes with sparse vegetation cover in regions of ephemeral snow packs?

How well does the measured accuracy and precision of $\Delta SD$ maps correspond to *a priori* estimates based on photogrammetric theory?

Our null hypothesis is that the accuracy of our approach will perform similarly to previous studies in terms of SD accuracy to as a function of vegetation and snowpack condition but will show lower bias when considering weekly $\Delta SD$ due to correlated errors related to surface conditions Further, we hypothesize that, except for very smooth snow pack conditions, the accuracy of $\Delta SD$ between snow covered dates will correspond to the expected accuracy from photogrammetric theory.

In Sect. 2 the study sites and methods used to estimate and validate $\Delta SD$ maps are described. A theoretical estimate of the precision of $\Delta SD$ as a function of mission parameters is also proposed. Results are presented in Sect. 3. Sect. 4 discusses these results in the context of the experimental conditions and their applicability to the research question. Conclusions with respect to the two research questions are given in Sect. 5.

## 2 Methods

### 2.1 Study Sites

Five study sites were located in two study regions: Gatineau and Acadia. To simplify the acquisition of permits for in-situ and UAV surveys, both study regions corresponded to land owned by the Government of Canada. The separation between regions was partly due to the availability of staff to perform surveys but also due to a desire to sample different snowpack and land surface conditions.



The Gatineau region (Figure 1) was located at 45°35' N latitude and 75°54' W longitude in Gatineau Park (a 391 km² federal park near Ottawa, Canada). The region consisted of land used for hay production with the meandering Meech Creek flowing across the southern half. Table 1 indicates recorded and climatological monthly rain, snow and temperatures for the nearest

5     climate station (Chelsea, Quebec at 45°31' N, 75°47' W, 112.50 m above sea level (a.s.l.)). During 2016, monthly air temperature was similar to the climate normal but rain (snow) was substantially higher (lower) than normal for March and lower (higher) for April. Two sites with alternatively flat and hilly macro-topography were established in the Gatineau region.

Gatineau North (GN) was a rectangular site of ~2.0 ha with grass cover less than 5 cm high over a flat surface. Gatineau South

10     (GS) was a rectangular site of ~3.2 ha centred on Meech Creek. The northern portion of GS (Figure 2) shared the same conditions as GN. The centre and southern portion of GS covered the river valley including spur hillslopes. Northern hillslopes where in-situ transects were located, were covered by low shrubs and grasses (<10 cm). Shrubs up to 1 m in height covered southern hillslopes. A small forested area was located at the South West corner of GS.

**Table 1. Monthly climate data from Chelsea, Quebec (Gatineau region). Normals correspond to 1981 to 2010.**

| MONTH | T (⁰C) | | Rain Fall (mm) | | Snow Fall (mm) | |
|---|---|---|---|---|---|---|
| | 2016 | Normal | 2016 | Normal | 2016 | Normal |
| JANUARY | -9.2 | -11.0 | 37.2 | 22.7 | 29.2 | 47.9 |
| FEBRUARY | -9.8 | -8.8 | 10.0 | 20.5 | 41.0 | 38.7 |
| MARCH | 1.9 | -3.0 | 106.9 | 34.6 | 5.2 | 26.5 |
| APRIL | 2.5 | 5.7 | 21.4 | 68.4 | 27.4 | 6.0 |
| MAY | 12.4 | 12.6 | 7.3 | 89.0 | 0 | 0 |

**Table 2. Monthly climate data for Fredericton, New Brunswick (Acadia Region). Normals correspond to 1981 to 2010.**

| MONTH | T (⁰C) | | Rain Fall (mm) | | Snow Fall (mm) | |
|---|---|---|---|---|---|---|
| | 2016 | Normal | 2016 | Normal | 2016 | Normal |
| JANUARY | -6.5 | -9.4 | 25.4 | 42.4 | 10.7 | 59.5 |
| FEBRUARY | -5.4 | -7.5 | 12.0 | 31.7 | 89.9 | 38.4 |
| MARCH | -2.1 | -2.2 | 7.2 | 45.2 | 82.0 | 44.9 |
| APRIL | 3.7 | 4.8 | 20.4 | 68.1 | 23.9 | 13.5 |
| MAY | 11.8 | 11.3 | 56.7 | 103.1 | 0 | 0.7 |







**Figure 1. Gatineau region showing Gatineau North (pink) and Gatineau South (blue) sites and ground control points (crosses) and in-situ transects (cyan lines).**

The Acadia region (Figure 2) was located at 45°58' N latitude and 66°19' W longitude in the Acadian Research Forest (a 91.6km² managed forest near Fredericton, Canada). The region consisted of three parcels of managed forest land, corresponding to sites Acadia A (AA), Acadia B (AB) and Acadia C (AC) respectively, separated by mature forest boundaries on gently undulating terrain. Table 2 indicates recorded and climatological monthly rain, snow and temperatures for the nearest

10     climate station (Fredericton, New Brunswick at 45°52'08" N, 66°32'14" W, 20.70 m a.s.l.). During 2016, monthly air





temperature was similar to the climate normal but rain (snow) was substantially lower (higher) than normal from February to April.

AA was a relatively flat trapezoidal site of ~3ha with grass (<5 cm) and stumps (<20 cm). AB was hummocky rectangular
5   site of ~4.5ha with stumps (<20 cm) and substantial brush and shrubs (<1 m) left over from clearing. AC was a rectangular site of ~4.5ha with recently planted Balsam Fir (Abies balsamea (L.) Mill.) ranging from 1 m to 5 m in height. AC was also hummocky although shrubs and herbs had covered most stumps. The sites were separated by mature mixed wood stands up to 20 m in height with balsam fir, red maple (Acer rubrum L.), and white birch (Betula papyrifera Marsh.).

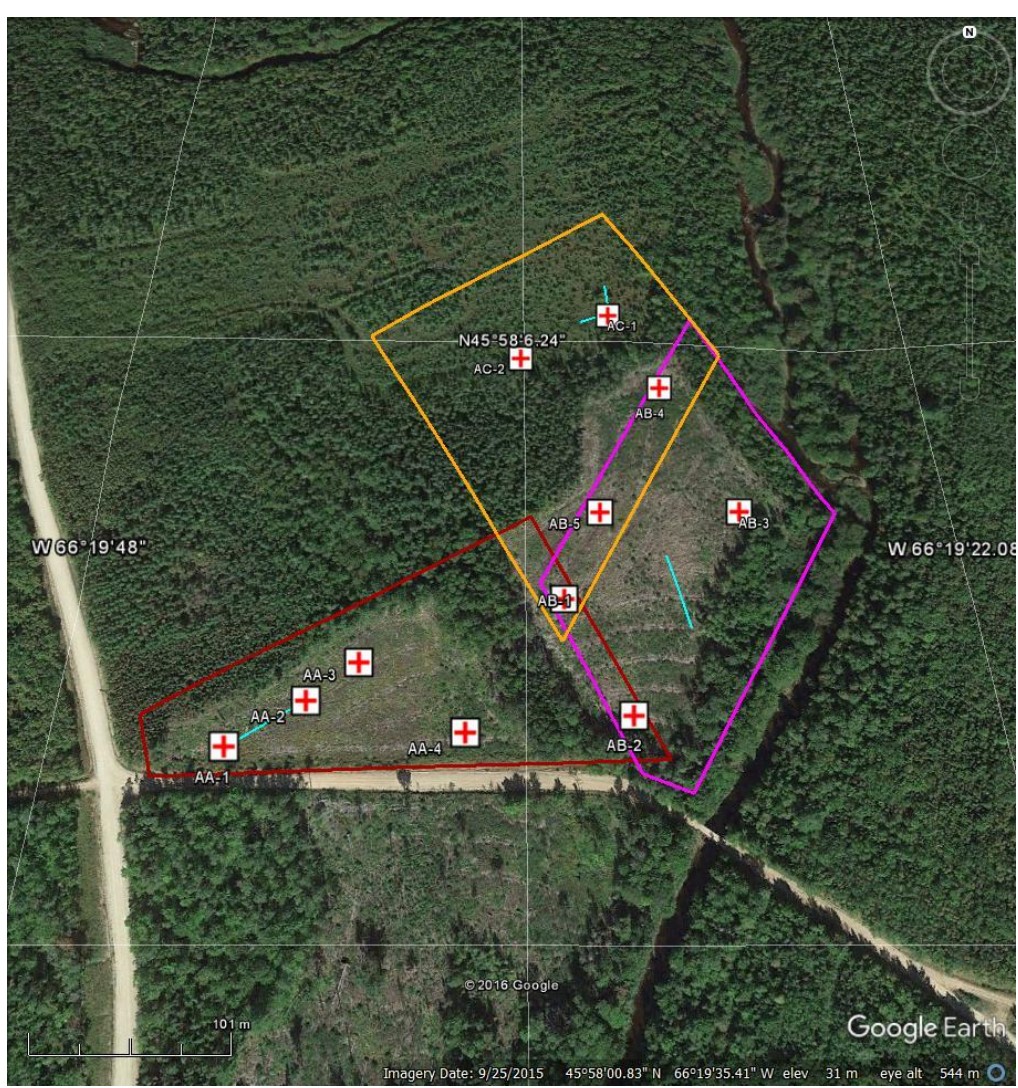

**Figure 2. Acadia region showing Acadia A (red), Acadia B (pink) and Acadia C (gold) sites with ground control points (crosses) and in-situ transects (cyan lines).**

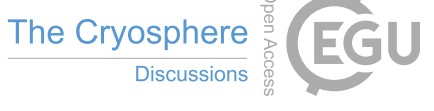

## 2.2 Ground Control Points

Ground Control Points (GCPs) were established for each site for geolocation of UAV imagery and derived maps.  The number
and location of GCPs were determined based on Tonkin and Midgley (2016) who assessed the accuracy of a digital surface
model (DSM) over a 14.5ha hummocky grassy landscape derived by applying SfM (Agisoft PhotoScan V1.1.5) to imagery
acquired using a Canon EOS-M 18 megapixel camera mounted on a hexacopter (altitude 100 m, forward speed 2 ms$^{-1}$, 95%
along track overlap, 75% across track overlap, 2 cm GSD).  They observed an average precision of 2.0 cm (83% circular error
probably or c.e.p.) in comparison to 530 check points.  Moreover, the average (extreme) vertical difference of the DSM was
not statistically different when the number of GCPs ranged from 4 (5) to 101.  They also observed a statistically significant
relationship between the vertical difference and the distance to the nearest GCP leading to their recommendation that GCPs
be ideally located within 100 m of mapped DSM locations.

Following the recommendations of Tonkin and Midgley (2016), at least 5 GCPs were positioned within the UAV coverage at
each site and at least one GCP near the corner of each site.  AC was exceptional as GCPs could not be located at the northern
edge due to access constraints. Six GCPs were located in GN and 10 GCPs in GS with 95% circular error probably of less than
2.05 cm (Prevost, 2016a).   For Acadia, four GCPs were located in AA, five GCPs in AB and two GCPs in AC with a 95%
circular error probable of less than 2.46 cm (Prevost, 2016b).

GCP targets at Gatineau consisted of both 30 cm square plywood (Figure 3a) and 15 cm diameter plastic disks (Figure 3b)
suspended between 1 m and 1.3 m above ground level on fence posts or poles.  The targets had a red background with a yellow
cross (for boards) or black centre (for disks) marked with tape.  Targets were cleaned prior to flights.   Based on experience at
Gatineau, GCP targets corresponding to plastic pylons, suspended on fence posts at ~1.3 m height (Figure 3c) were used at
Acadia to reduce the need to clear snow from targets and to assist in identifying the centre of the GCP target within UAV
imagery.  Black tape was used to mark vertical stripes on the cones to increase their visibility.



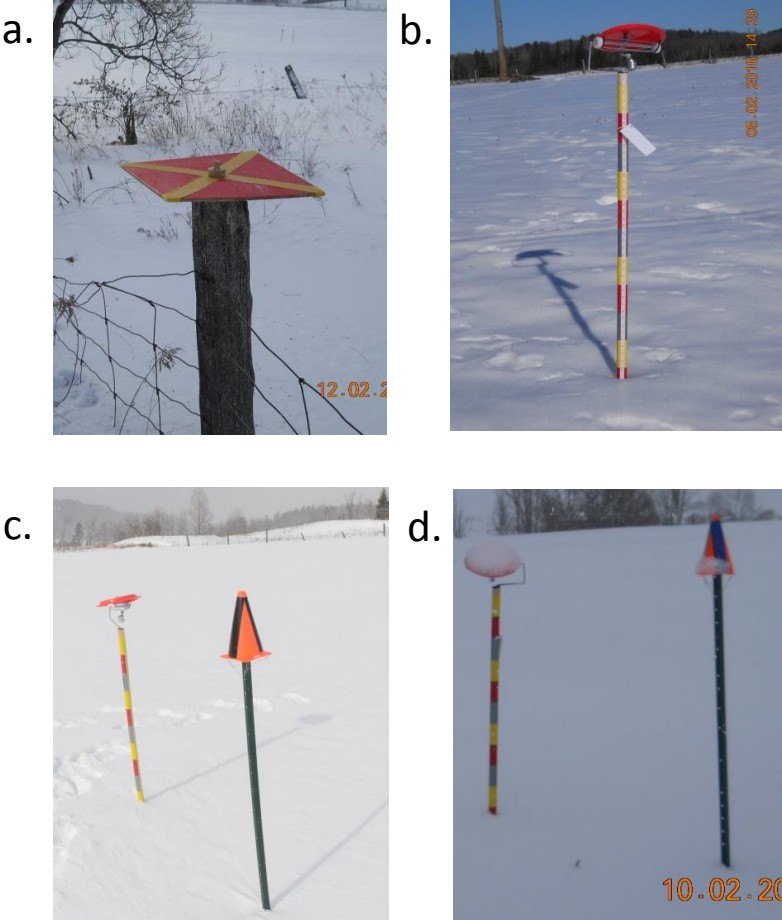

**Figure 3. GCP Targets: a) square plywood b) disk on pole and c) disk on pole and cone on pole snow free d) disk on pole and cone on pole with snow.**

## 5    2.3 In-Situ $\Delta SD$ Measurement

Transects of ~50 m length (see Figures 1 and 2) were positioned at each site within 5 m of a GCP. Each site had one transect except for GS where two transects were located (GS-1 in the flat northern portion and GS-2 along and across a spur hillslope leading into the floodplain). Along each transect, twelve 48" x 2" x 1" wooden stakes were placed ~10 cm deep and
10   approximately vertical and equally spaced apart. Stakes were covered with black all weather tape in addition to two red bands each 10 cm wide separated by 50 cm (Figure 4). The attitude of the stakes was measured at the start and end of the field



season using a digital level to a precision of $0.1^0$ . The elevation of the stakes above the soil layer was measured at the end of the field season using a plumb line and tape measure to a precision of better than +/-0.5 cm (95% confidence interval).

In-situ $\Delta SD$ was estimated at each stake using the protocol described in Oakes et al. (2016). For snow free conditions, the
freeboard (F), defined as the stake height above the current surface, was determined from the plum-bob measurement. Otherwise, F was determined using an in-situ high resolution digital image. For each stake, a 14 Mpixel photograph (Nikon D7000 camera and 70-300 mm / f4.5 Nikon lens) was taken ~5 m parallel to the transect using manual focus and automatic exposure. To reduce precision errors due to localized snow melt or drifting at the stake, the point of intersection of the snow pack and the stake was visually determined by interpolating the snow pack horizon closest to the front of the stake (e.g. Figure
4) rather than within the well (or mound) of snow adjacent to the stake. The distance from the top of the stake to the edge of each visible red-tape band and to the midpoint of the snow pack intersection with the stake was measured in pixel units using Adobe Photoshop. Freeboard was then estimated using the ratio of distances in pixel units and the known distance between bands and converted to a vertical distance using measurements of the stake angle. The difference in F between two dates was used to estimate $\Delta SD$ at each stake. When comparing snow covered conditions, the uncertainty for measuring the $\Delta SD$
assuming independent errors in determining F is ~2.06 cm (95% confidence interval) for typical uncertainties in delineating F and the stake angle (Oakes et al., 2016). As both sources of uncertainty are spatially random the uncertainty in estimating the average snow depth using all 12 stakes in a transect is ~0.60 cm (95% confidence interval).

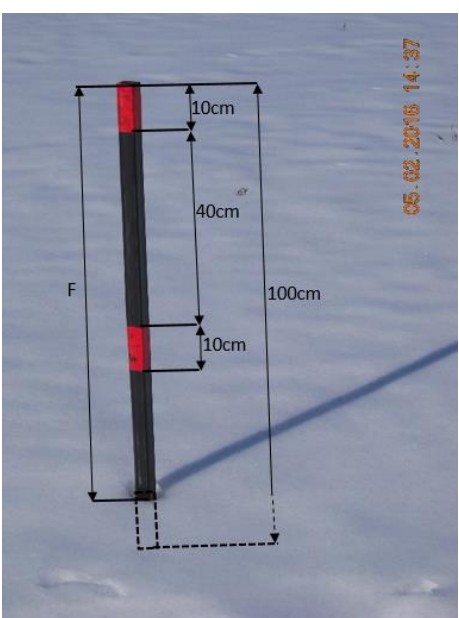

**Figure 4. In-situ snow stake measurements. Dashed lines correspond to locations below the snow surface.**





## 2.4 UAV Missions

Missions were performed weekly at Gatineau (26/01/2016 to 19/04/2016) and Acadia (10/02/2016 to 14/04/2016) during periods without precipitation at the start of the mission using a Phantom Pro 3 Plus UAV (https://www.dji.com/phantom-3-pro; P3P).  The same UAV was used for all missions in a region.   Imagery was acquired using the provided gimbal mounted 12.4 Mpixel rolling-shutter camera with a f2.8 fixed aperture in auto-exposure mode recording in 4K MPEG-4 AVC/H.264 format (MP4) (Table 3).  MP4 was used in preference to full resolution photographs since i) the system firmware limited the maximum photograph sampling rate to 1 frames/2s and ii) the 4K video frames have almost identical resolution to the full resolution photographs (Leblanc 2018).   Auto-exposure mode was used since the flights encountered rapid variations between sunlit and shaded snow and vegetation areas making it challenging to adjust exposure manually during the flight.

**Table 3.  UAV Specifications**

| PARAMETER | VALUE | ABBREVIATION |
|---|---|---|
| MASS | 1280g | - |
| OPERATING TEMPERATURE | 0 $^0$C to 40 $^0$C | $T$ |
| FLIGHT TIME PER BATTERY | 23 minutes | $t_{max}$ |
| MAXIMUM CRUISING SPEED | 16 m/s | $v$ |
| VERTICAL PRECISION | 0.5 m | $\delta_z$ |
| HORIZONTAL PRECISION | 1.5 m | $\delta_x$ |
| LENS FOCAL LENGTH | 3.66624 mm | $c$ |
| CAMERA APERTURE | f2.8 | $F$ |
| DIAGONAL FIELD OF VIEW | 94$^0$ | $\theta$ |
| CAMERA SENSOR | Sony Exmor IMX377 | - |
| DETECTOR SIZE | 1.55 ☐m | $l$ |
| #VERTICAL PIXELS | 3044 pixels | - |
| #HORIZONTAL PIXELS | 4072 pixels | - |
| VIDEO FRAME RATE | 24 frames/s | $f_{max}$ |
| VIDEO VERTICAL RESOLUTION | 2160 pixels | $n_y$ |
| VIDEO HORIZONTAL RESOLUTION | 4096 pixels | $n_x$ |
| VIDEO EFFECTIVE DETECTOR SIZE | 1.57937 cm | $l_e$ |





Lichee V3.0.4 (https://flylitchi.com/new) flight planning software was used to create flight plans. The same flight plan was used for all missions at a site. Flight plans were defined using equally spaced parallel linear tracks flying oriented North to ensure consistent locations of shadows between dates. The exception was AC where tracks were oriented parallel to the GPS targets at AB to maximize overlap over these targets. Cross tracks were not used since this would increase flight time and

since Nasurallah (2016) found that they did not significantly improve point cloud accuracy or density when using data acquired using a similar consumer grade UAV and SfM software. Flight plans were defined to cover rectangular (triangular in the case of AA) regions with a buffer of 100 m to ensure adequate side views at the edges of each study area and to include GCPs from adjacent sites. Flights were planned such that the UAV was always flying along the vertical axis of the camera to minimize post processing complexity. Turns were limited to 90° with smoothing of arcs to provide adequate side overlap.

For convenience, missions were constrained to a single P3P battery. Since surveys were to be conducted in cold and windy conditions a maximum flight time ($t_{max}$) of 17.25 minutes was used for flight planning. The effective time for image acquisition ($t_{eff}$) was further reduced to 15 minutes to accommodate travel time to and from the launch location and to execute turns between flight tracks. Mission parameters were optimized to minimize the vertical precision error in altitude $H$ ($\sigma_H$)

derived from the block triangulation of images at matching key points covering a nominal mapped extent of 10ha. For a matching key point found in $K$ images each acquired at a lateral distance of $d_k$ from the key point (Forstner, 1998):

$$\sigma_H = \frac{H^2}{c} \frac{\sigma_x \sqrt{12}}{\sqrt{\sum_{k=1}^{K} d_k}} \tag{1}$$

where $\sigma_x$ is the average horizontal uncertainty when matching the location in each image pair on the camera focal plane and $c$ is the lens focal length. Ignoring edges of flight tracks, $\{d_k\}$ and therefore $\sigma_H$ will be a function of the along track image spacing ($b_y$), the across track image spacing ($b_x$) and $H$. With 4K video it is generally possible to chose a frame sampling rate $f$ such that $b_y \leq b_x$.

Eq. 1 assumes that matches are found in all overlapping images. Based strictly on geometric considerations, for the P3P with $H \leq 100m$ and $b_x < 40$ m we have $K > 20$ matches. In practice, $K$ is much less than 20 due to the difficulty in matching the same feature in multiple images (Nasrullah, 2016). Adopting the worst case assumption that the matched images are closest to the key point location and assuming similar along and across track spacing, from Forstner (1998):

$$\sigma_H \leq \frac{H^2}{c b_x} \frac{\sigma_x \sqrt{12}}{\sqrt{K(K^2 - 1)}} \tag{2}$$





Here, $\sigma_x$ was estimated as the Euclidean sum of the mean reprojection error after block adjustment $\sigma_{re}$, the uncorrected motion blur during integration of the detector signal ($\sigma_m$), and the uncorrected rolling shutter motion ($\sigma_{rs}$).

$$\sigma_x^2 = \sigma_{re}^2 + \sigma_m^2 + \sigma_{rs}^2 \tag{3}$$

Mean reprojection error is computed during block adjustment by the Pix4D Mapper Pro. Motion blur is given by

$$\sigma_m = \frac{v_y c \tau_e}{Hl} \tag{4}$$

10  where $v_y$ is the along track velocity, $\tau_e$ is the exposure time and $l$ is the detector size along track. Rolling shutter correction error is determined by the uncertainty in $v$ and the sensor readout time $\tau_s$:

$$\sigma_{rs} = \sigma_v \frac{c \tau_s}{Hl} \tag{5}$$

15  Estimates of $K$, $\sigma_{re}$, $\sigma_m$ and $\sigma_{rs}$ were required to optimize $\sigma_H$ with respect to flight parameters. Trial flights using parameters given in Table 4 were performed at both GN and GS on one sunny (January 26, 2016) and one overcast (February 2, 2017) day with complete snow cover and processed using Pix4D Version 3.0. The lowest feasible $H$ of 50 m (to ensure clearance of terrain and cover a 10 ha site using one battery) was selected to provide a best case estimate of $K$ corresponding to the smallest feasible GSD.

**Table 4. Mission parameters for all flights. The nominal 10ha study area assumes rectangular region with 300 m transects.**

| Parameter | Value | Abbreviation |
|---|---|---|
| Height | 50 m | $H$ |
| Speed | 15 m/s | $v$ |
| Ground Sampling Distance | 0.021 m | $GSD$ |
| Effective Shutter Speed | <0.02 s | $\tau_e$ |
| Motion Blur | 0.039 pixels | None |
| Track spacing | 15 m | $b_{ac}$ |
| Frame sampling interval | 1 s | None |
| Across Track Overlap | 82% | None |
| Along Track Overlap | 93% | None |
| Minimum study area | 10 ha | $A$ |




Figure 5 indicates that $K$ followed an exponential distribution that was relatively consistent over the four flights. Keypoints with $K = 2$ matches are discarded as insufficiently accurate to include in the $\sigma_x$ estimation. In this case, the average $K$ over the four missions was 5.5 matches with a range of 4.3 matches to 7.4 matches. The two overcast dates had lower than average

5  $K$ while the sunny dates were above average. These values of $K$ are substantially lower than the maximum possible $K$ based only on geometric considerations but are similar to values reported in Nasrullah (2016).

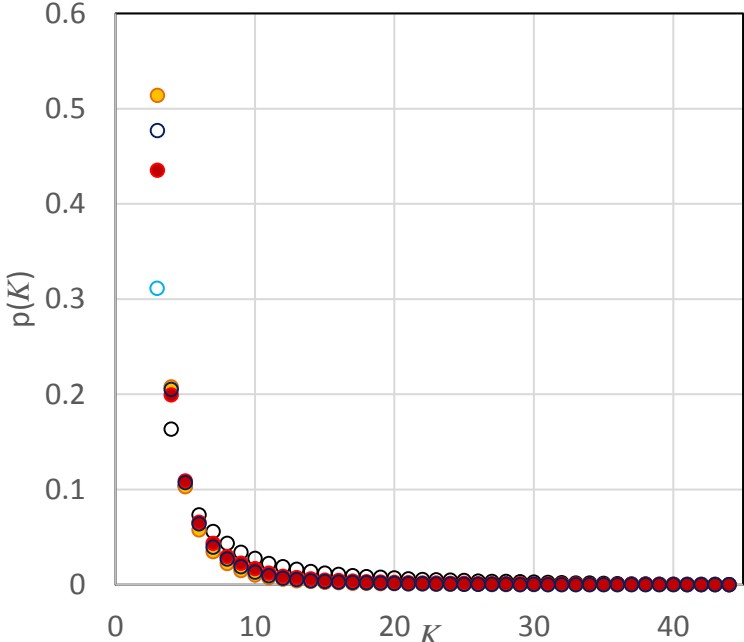

**Figure 5. Empirical probability of observing $K$ matches for points acquired during four trial missions (filled symbols are for overcast dates)**

For the four test flights $\sigma_{re}$ ranged of 0.179 pixels to 0.209 pixels, and $\tau_e$ ranged from 0.017 s to 0.005 s. Worst case values of $\sigma_{re} = 0.25$ pixels and $\tau_e = 0.02$ s were used for selecting flight parameters. We did not have sufficiently accurate on-board sensors to provide reference values of $\sigma_v$. Instead, we relied on a published comparison of $v$ based on imagery from a PIX4D block adjustment and on-board measurement (Vautherin et al., 2016) indicating $\sigma_v \approx 0.05\, v$.

Using measurements from the training flights the relationship between $\sigma_H$ and $H$ was modelled for the average and extreme values of $K$ using the 10ha minimum area constraint to relate $v_y$ to $b_x$. Figure 6 indicates that $\sigma_H$ increases almost linearly with $H$ for any given $K$ although the rate of increase is steeper for low $K$. This result indicates it is critical to select the lowest feasible $H$. At $H = 50\, m$ the sensitivity of $\sigma_H$ to $b_x$ is negligible (<10% $\sigma_H$) for $15\, m \leq b_x \leq 30\, m$. Here we selected $b_x =$

20  $15\, m$ to maximize across track overlap since we were able to increase $f$ to achieve a constant along track overlap irrespective





of $b_x$. This was important since the density of key points increases with overlap with all other parameters fixed (Nasrullah 2016). The selected flight parameters predict a $\sigma_H = 1.44\ cm$ for $K = 5$ matches (ranging from $\sigma_H = 0.92\ cm$ for $K = 8$ matches to $\sigma_H = 3.73\ cm$ for $K = 3$) matches. As $\Delta SD$ was estimated by computing the temporal difference of DSMs (Sect. 2.7) its precision error, assuming uncorrelated precision errors in $H$ between two dates, corresponds to the Euclidean sum of

5   $\sigma_H$ for each date. Ignoring uncertainty due to surface roughness for snow free conditions, $\sigma_{\Delta SD} = 1.2\ cm$ where both dates have $K = 8$ matches, $\sigma_{\Delta SD} = 2.14\ cm$ where both dates have $K = 5$ matches and a worst case $\sigma_{\Delta SD} = 5.25\ cm$ where both dates have $K = 3$ matches.

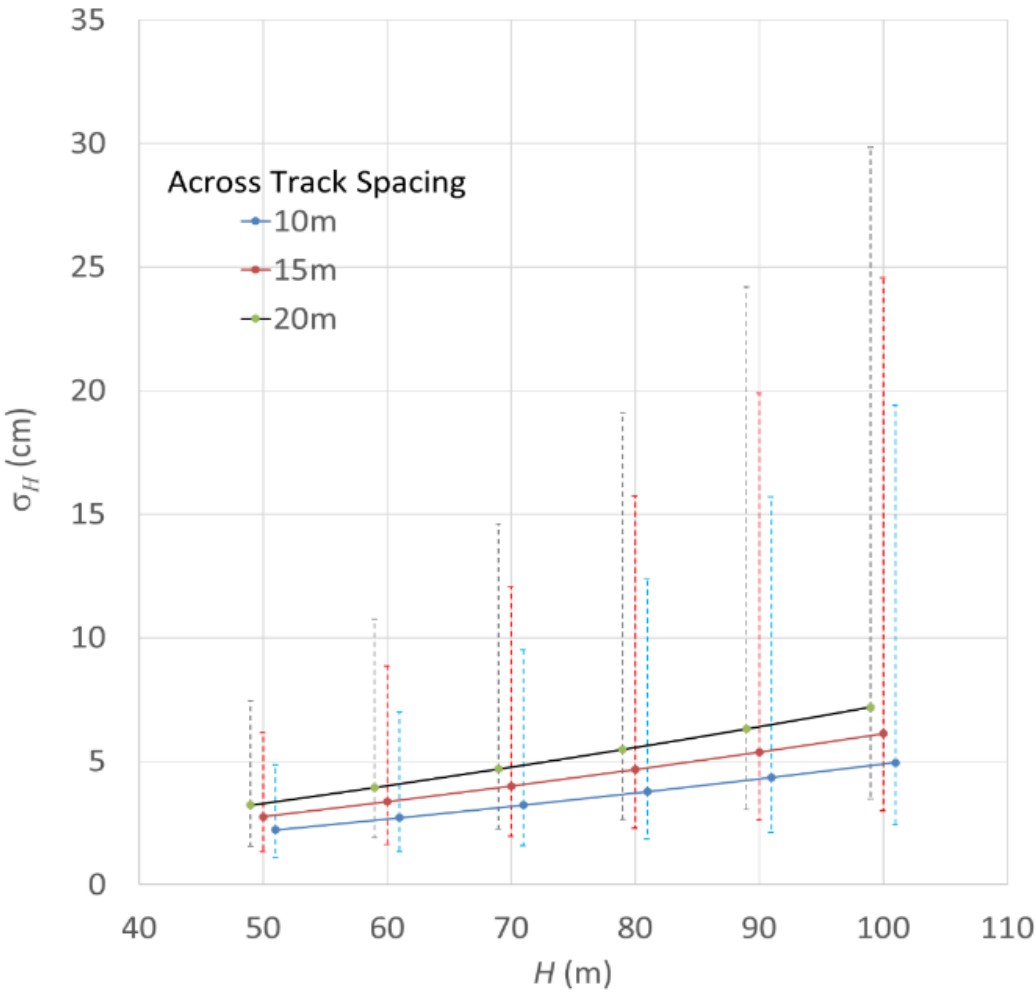

10   **Figure 6. Theoretical relationship between vertical uncertainty and UAV height. Solid lines correspond to 5 matching images per point. Upper (lower) bars correspond to 3(8) matching images per point. Black and blue lines as horizontally offset from red line for clarity.**



## 2.5     UAV Video Processing

Each UAV mission resulted in two consecutive MP4 videos (due to a limitation of 3.91Gbytes for a single MP4 file) and an ephemeris file providing the P3P position and attitude with a temporal resolution of about 0.1s.  Data from each mission was

processed in Pix4D Mapper PRO Version3.2 (https://pix4d.com/product/pix4dmapper-photogrammetry-software/) as follows:

I.     The two videos were subsampled with a 1s interval and extracted as JPEG images together with ephemeris data to provide an initial location for each image.  The 1s interval ensured the desired along track overlap while minimizing computation.

II.     Nominal geolocation uncertainty was specified as 5 m horizontal and 10 m vertical considering that the P3P was always operating within Wide Area Augmentation System coverage.

III.     Camera parameters were initialized using the P3P Video specification with rolling shutter.

IV.     An initial camera calibration and point cloud was produced using the Pix4D algorithm for feature matching and bundle adjustment with rolling shutter correction.

V.     Each GCP was manually geolocated in as many (at least 10) JPEG images as feasible.

VI.     Feature matching was repeated and internal and external camera parameters refined using bundle adjustment with rolling shutter correction at the highest quality setting.

VII.     If the GCPs were not fit within 5 cm root mean square difference (RMSD) steps V. and VI. were repeated once.

VIII.     The point cloud was densified using the Pix4D default two pixel sub-sampling of images.

IX.     The resulting dense point cloud (PC) was exported to MATLAB in XYZ format.

X.     To quantify the geolocation uncertainty of the point cloud the internal and external camera parameters were refined while holding out individual GCPs.




## 2.6 Assessment of Micro-topography

Micro-topography was assessed for each transect within each site using a snow free PC acquired within one week of complete snowmelt. Compressed vegetation was included within micro-topography since it also acts to bias estimates of $SD$ (Harding
et al., 2016). Micro-topography was quantified as the RMSD from a local robust linear slope trend (MATLAB function 'lmfit' with robust option, https://www.mathworks.com/help/stats/fitlm.html) with a 15 m moving window oriented along the transect. Positive deviations greater than the maximum snowpack elevation at each transect during the season were removed when computing the RMSD over a transect to eliminate overstory vegetation that normally would be above the snowpack.

## 2.7 Elevation and Overstory Cover Extraction

Surface elevations were extracted from each PC in a sampling region around each stake. The sampling region was held constant for all missions over a given site. The sampling region corresponded to a 2 m tall vertical elliptic cylinder centred on the nominal horizontal location of a stake and extended 50 cm below the nominal vertical location of a stake. The horizontal
(vertical) centre of the sampling region was specified as the average (average less 50 cm) of the visually determined location of the base of the stake from the colorized PCs for two missions acquired during sunny conditions with less than 5 cm snow depth. The 50 cm vertical offset was required to account for both PC geolocation uncertainty and local topography (including snow pits due to melt at the based of the stake) close to the stake. The horizontal major and minor axes of the cylinder were specified to approximate twice the Euclidean sum of geolocation uncertainty of the PC and the typical geolocation uncertainty
of the stake corresponding to the difference between both reference image locations. These considerations typically resulted in horizontal axes lengths ranging from 10 cm to 24 cm depending on the precision of the stake geolocation between reference images.

The average overstory vegetation cover in the vicinity of transect sampling locations was estimated for each UAV mission.
Overstory vegetation cover near each stake was estimated for a 1 m radius cylinder centred horizontally at each nominal stake location as the fraction of grid cells where at least one other point was found vertically above a surface point. A 1 m radius was used as an approximation of points within the field of view of images used to map the elevation in the smaller region used around each stake.



## 2.8        Δ*SD* Estimation from Point Clouds

Δ*SD* was estimated for each transect using geolocated PCs.  For each PC, snow cover points were identified in each sampling window using points exceeding the 50%ile of the blue band in a sampling region.  The blue band was used as a simple indication of snow considering that vegetation and shadows should both have substantially lower blue intensity in a region with similar view geometry and similar top of canopy illumination conditions (Miller et al., 1997).  To minimize bias due to the presence of melt depressions at the base of each stake and due to snow on vegetation, the median elevation of snow cover points was used to represent the snow surface elevation at each stake.

Snow free surface elevation was estimated as the median elevation of all points falling in the sampling window, that were unobstructed by points vertically below them, for a PC produced using a UAV flight over snow-free conditions within one week after complete snow melt.  For each UAV flight, the average Δ*SD* across all sampling windows for the transect was used to estimate the transect Δ*SD*.  The precision of Δ*SD* was estimated using the central 67.5%ile interval of sampled Δ*SD* within the transect to include both measurement error and natural variability.

## 2.9 Performance Assessments

The performance of geolocated DSMs and Δ*SD*, in comparison to reference values from GCPs and in-situ transects respectively, was reported in terms of accuracy, precision and uncertainty statistics following ANSI/NCSL (1997).  Here accuracy is defined as the mean difference between sampled validated and reference data (i.e. the bias), precision is the RMSD after subtracting the accuracy from the validated data, and uncertainty is the RMSD between the validated and reference data. For convenience we use the term 'bias' for accuracy and 'RMSD' for uncertainty. In contrast to previous studies that report RMSD in comparison to individual in-situ sample locations, assessments were performed using transect averages since addressing the broader research goal of combining in-situ and UAV based Δ*SD* requires an assessment of UAV estimates of Δ*SD* over a sampling footprint comparable to the reported in-situ measurement (i.e. transect average at ruler locations).

Camera calibration performance was assessed in terms of the percentage of images (*P*) successfully calibrated using a single block adjustment, the number of keypoint matches per image, and the density of keypoint matches per square metre (D).



## 3.0 Results

### 3.1 Data Acquisition

5  UAV flights were conducted on 13 days at Gatineau and 16 days at Acadia resulting in 74 missions.   For brevity, results for a mission are referenced using the site acronym followed by the date (e.g. GS 26/01/2016 is the Gatineau south mission for 26/01/2016).  Flights were performed between 10:00 and 14:00 local time.  Environmental conditions for each date are provided in Tables 5 and 6 based on the nearest climate station.  Maximum daily temperatures at Gatineau (Acadia) ranged from -7.6 °C (-9.0 °C) to 14.5 °C (11.5 °C) and can be considered representative of typical temperature variability during late

10 winter and spring melt periods.  Average wind speed ranged from 3 m/s to 26 m/s although the higher value may not be representative of local conditions since flights were not conducted if there was strong evidence of surface gusts or swaying conifer trunks.  Sky conditions included both cloudy and overcast with one instance (GN 10/02/2016) where snow was falling. Snowpack conditions included fresh snow, icy snow, wet snow, patchy snow (incomplete cover) and snow free.  Ephemeral melt, preceded by over 10 mm of rain, occurred at both Gatineau (02/02/2016) and Acadia (18/02/2019).

**Table 5.  Environmental conditions during P3P missions over Gatineau.  Rain and snow correspond to cumulated totals since previous mission.  Melt periods are in bold font.**

| DATE | $T_{MAX}$ °C | CUM. RAIN MM | CUM. SNOW CM | WIND SPEED MS$^{-1}$ | SKY CONDITIONS | SNOW CONDITIONS |
|---|---|---|---|---|---|---|
| 2016-01-26 | -5.5 | No data | No data | 12 | Clear | Icy |
| 2016-02-02 | 0.5 | 10.8 | 1.6 | 3 | Clear | Wet |
| 2016-02-10 | -3.5 | 0 | 6.4 | 4 | Snowing | Dry |
| 2016-02-12 | -5.5 | 0 | 2.0 | 10 | Overcast | Fresh Snow |
| 2016-02-17 | 0.5 | 0 | 23.6 | 3 | Overcast | Icy |
| 2016-02-18 | -6.5 | 0 | 0.6 | 7 | Clear | Dry |
| 2016-02-22 | -7.6 | 10.0 | 5.4 | 11 | Clear | Icy |
| 2016-02-29 | 0.1 | 27.4 | 9.2 | 9 | Overcast | Dry |
| 2016-03-04 | -6.0 | 0 | 9.1 | 10 | Clear | Icy |
| **2016-03-17** | **7** | **33.4** | **0** | **6** | **Clear** | **Wet** |
| **2016-03-21** | **4** | **0** | **0** | **18** | **Clear** | **Wet** |
| **2016-03-26** | **3.8** | **11.0** | **5.2** | **11** | **Clear** | **Wet** |
| 2016-04-19 | 14.5 | 72.4 | 0 | 22 | Clear | Bare |



Three missions were not processed due to issues with the recorded data. In one case (GS 26/01/2016) the camera was pointed horizontally rather than nadir looking down. In the other two cases (AA 23/02/2016 and AC 10/03/2016) the mission was aborted due to a communication error between the flight controller and the UAV. It was later determined this error was due to a conflict between automatic updates of the Lichee software and manual updates of the P3P control software.

**Table 6. Environmental conditions during P3P missions over Acadia. Rain and snow correspond to cumulated totals since previous mission. Melt periods are in bold font.**

| DATE | $T_{MAX}$ $^0$C | CUM. RAIN MM | CUM. SNOW CM | WIND SPEED MS$^{-1}$ | SKY CONDITIONS | SNOW CONDITIONS |
|---|---|---|---|---|---|---|
| 2106-02-10 | -3.0 | 0 | 22.8 | 9 | Clear | Dry |
| **2016-02-18** | **0** | **26.1** | **1.6** | **11** | **Clear** | **Wet** |
| **2016-02-19** | **-1.0** | **22.0** | **6.0** | **26** | **Clear** | **Wet** |
| **2016-02-23** | **-8.0** | **5.4** | **0** | **10** | **Clear** | **Wet,Patchy** |
| **2016-03-04** | **-3.5** | **50.0** | **0** | **21** | **Clear** | **Wet,Patchy** |
| **2016-03-06** | **-2.0** | **0** | **0** | **25** | **Clear** | **Wet,Patchy** |
| **2016-03-08** | **4.0** | **0** | **2.0** | **24** | **Clear** | **Wet,Patchy** |
| **2016-03-10** | **4.0** | **5.5** | **0** | **18** | **Overcast** | **Wet,Patchy** |
| 2016-03-11 | 0 | 0 | 4.3 | 11 | Overcast | Dry |
| 2016-03-14 | 0 | 0 | 0 | 13 | Clear | Dry, Patchy |
| 2016-03-20 | -9.0 | 0 | 0 | 13 | Clear | Dry |
| 2016-03-23 | -1.0 | 19.0 | 0 | 18 | Overcast | Icy |
| 2016-03-24 | -8.0 | 0.5 | 12.2 | 5 | Overcast | Fresh Snow |
| 2016-03-26 | -2.0 | 11.2 | 4.0 | 4 | Clear | Dry |
| **2016-03-30** | **3.0** | **39.3** | **0** | **13** | **Clear** | **Fresh Snow** |
| 2016-04-14 | 11.1 | 65.8 | 0 | 15 | Overcast | Bare |

## 3.2 UAV Data Processing

Seventy-one missions were processed with Pix4D (details in Supplementary Material). Of these, three missions over GN, corresponding to either snowing or icy snow conditions, resulted in <500 matches/image and subsequently $< 50\%$. Two other missions (GN 29/02/2016) and (AB 08/03/2016) also resulted in less than 50% images with successfully calibrated virtual cameras. During both of these missions, there was spatially uniform fresh snow that possibly reduced the number of





spatial features suitable for matching. The remainder of the missions were each processed using a single block adjustment with a median $P = 97\%$ (minimum $= 80\%$).

The key point match density varied significantly between missions and sites (Figure 7). Fresh snow or ice conditions resulted
in $D < 10$ $m$atches/m$^2$ irrespective of the site. Season average $D$ was higher over Acadia (83 matches/m$^2$) than Gatineau (28 $m$atches/m$^2$) even considering only dates without icy or fresh snow (91 $m$atches/m$^2$ for Acadia versus 42 matches/m$^2$ for Gatineau). For dates at or exceeding the median $D$, $K$ ranged from 4 to 8 (not shown). Pix4D does not provide a similar statistic over sub-areas. Missions with differing sky conditions but constant snowpack conditions only occurred at AA for one pair of dates (08/03/2016 and 10/03/2016) when missions were repeated due to instrument failure on the first date at AB. For
these two missions, $D$ was higher under clear versus overcast conditions but there was insufficient replication to determine if this impacted $\Delta SD$ estimation.

Horizontal uncertainty ranged from 0.44 cm to 11 cm with a median of 1.87 cm while the vertical uncertainty ranged from 0.045 cm to 4.6 cm with a median of 1.02 cm (Figure 8). Over 75% of missions resulted in a geolocation uncertainty under 4
cm in both horizontal and vertical. Uncertainty less than 0.5 cm RMSD was only observed for missions with $D > 50$ matches/m$^2$ but uncertainty was unrelated to $D$ past this matching density. Horizontal accuracy ranged from -0.68 cm to 0.57 cm (median -0.01 cm) and vertical accuracy ranged from -1.10 cm to 0.48 cm (median -0.04 cm) (Figure 9). There was evidence of a linear relationship between vertical and horizontal accuracy after accounting for outliers. Horizontal precision ranged from 0.04 cm to 10.7 cm (median 1.76 cm) and vertical precision ranged from 0.04 cm to 4.5 cm (median 0.99 cm)
(not shown). A least absolute residual regression of vertical versus horizontal precision gave an adjusted r$^2$ of 0.97 with a slope of 1.11 (95% confidence interval [1.04,1.18]). Precision error was closely related to uncertainty due to the low bias relative to uncertainty (not shown). Similar to accuracy, a least absolute residual regression of vertical versus horizontal precision gave an adjusted r$^2$ of 0.95 with a slope of 0.58 (95% confidence interval [0.54, 0.62]).

As expected, micro-topographic roughness increased from qualitatively smooth to rough sites with values under 5 cm at Gatineau, between 5 cm and 10 cm at AA and AB and 42 cm at AC (Figure 10). AC indicated the presence of high spatial frequency variation (length scales < 10 cm) that were due to low vegetation rather than variations in ground surface elevation *per se*.





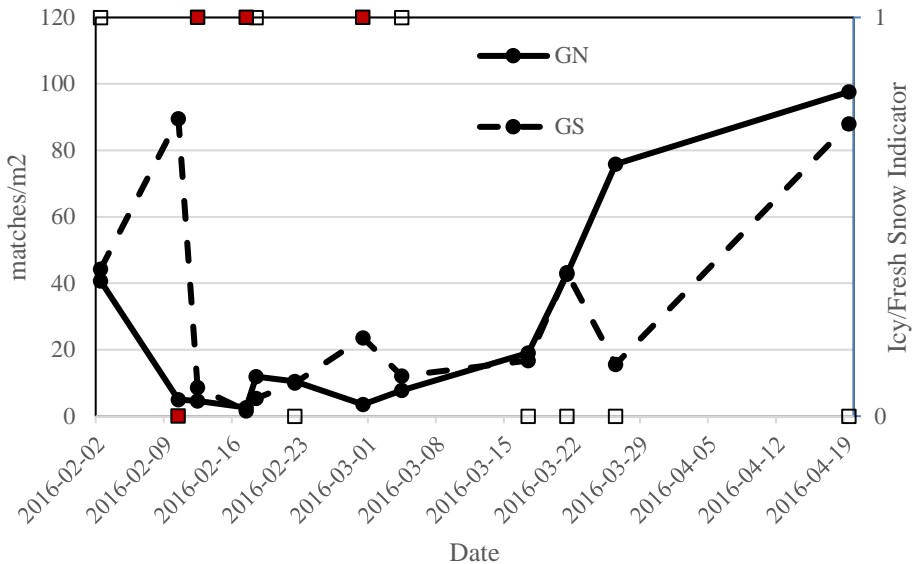

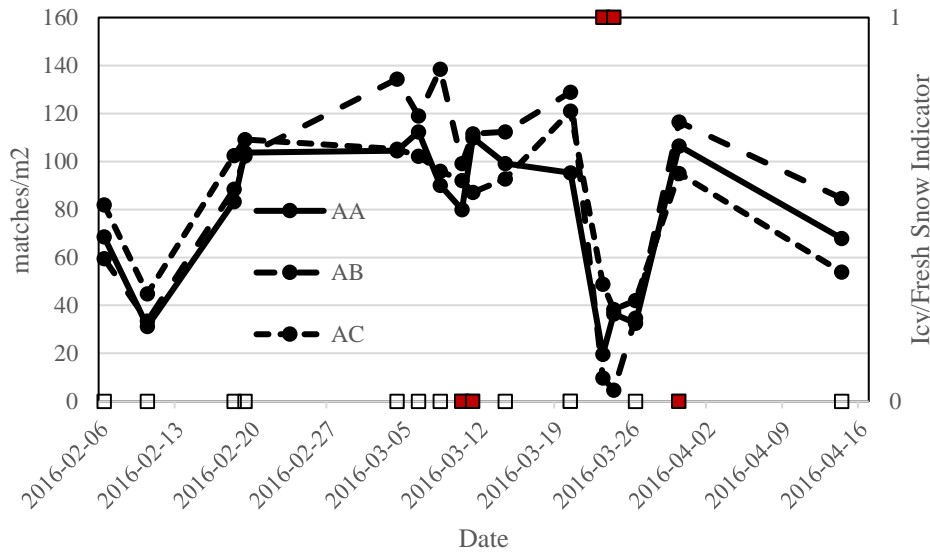

**Figure 7. Pix4D automated tie point match density for Gatineau (upper panel) and Acadia (lower panel) together with indicator of fresh snow (square symbols). Missions (solid circular symbols) for the same site are connected by lines. Red squares indicated overcast conditions.**


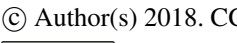

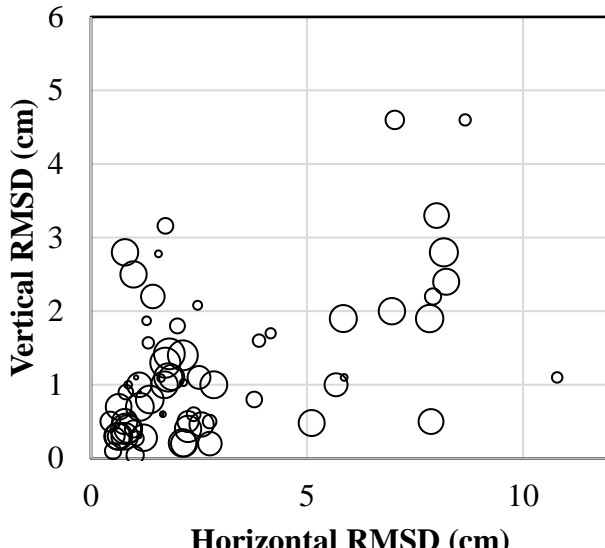

**Figure 8.** Uncertainty of absolute geolocation for digital surface models based on cross-validation with GCPs.

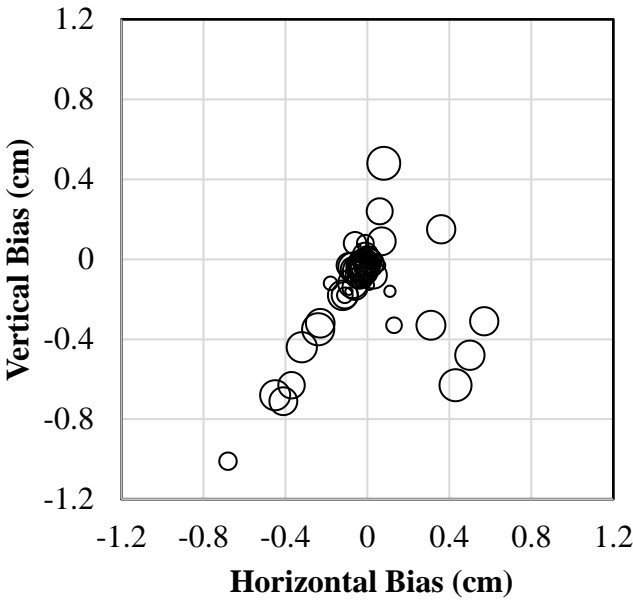

**Figure 9.** Accuracy of absolute geolocation for digital surface models based on cross-validation with GCPs.



**Figure 10.** Deviations from local robust linear trend (based on 15 m moving window) of PC elevations along each transect. Only the first 15 m of each transect are shown for clarity. AC is truncated as the transect consisted of shorter line segments.



## 3.2 Δ*SD* Mapping Performance

The performance of Δ*SD* mapping was evaluated in terms of both changes between successive dates and changes between a given date and snow free conditions. Figures 11 shows the Δ*SD* between successive dates for each transect. Vertical (horizontal) bars indicating their one standard deviation confidence interval due to within transect variation in Δ*SD* from the image data (in-situ data). The bars indicate that spatial variability in Δ*SD* within a transect was often larger than the 0.30 cm (1 standard deviation) uncertainty for in-situ Δ*SD* estimation for a transect assuming no spatial variability. As such, the in-situ measurement method was considered sufficiently precise for reference estimates. Nevertheless, due to within transect variation in Δ*SD*, the precision of both in-situ and image based methods was often similar in magnitude to observed Δ*SD* so that statistically significant comparisons could not be conducted for individual dates. Rather, in-situ and image based Δ*SD* were compared using statistics based on differences observed for all dates for each transect. In this case, uncertainty ranged from 2.54 cm to 5.12 cm for the non-forested sites to 8.68 cm at AC. The temporal bias was substantially smaller than uncertainty, ranging from between -0.80 cm at GS to 0.35 cm at AC. As such the precision error was only slightly less than the uncertainty (not shown). There were seven instances where the observed difference exceeded 5cm. Four were overestimates ranging from 5cm to 10cm at Gatineau and the other three were all at AA including the largest residual corresponding to an underestimate of 20cm. All of these cases involved estimating change ivolving at least one date with either extremely icy snow and another with deep fresh snow. In such cases the PC density can be low (Figure 7) while the snowpack itself has changed substantially between dates. Moreover, for AC, the identified key points were often at snow-vegetation intersections (not shown) that may differ systematically in Δ*SD* when compared to the stakes that were placed within openings.

Figure 12 compares Δ*SD* between snow covered and snow free conditions (i.e. estimated *SD*). In this case, the confidence interval of Δ*SD* for a transect is on average +5.2 cm/-6.3 cm for in-situ and +4.1 cm/-7.8 cm for image based estimates. Uncertainty ranged from 1.58 cm at AB to 10.56 cm at GN. Accuracy varied between sites. Bias was below 1.2cm for GS T1, AA and AB. In contrast, bias at the other sites exceeded +/-5cm (-10.05cm at GN T1, -6.23cm at GS T1 and 5.5cm at AC). Moreover, the bias was consistent over time with the exception of large (>5cm) under estimates for the date just prior to snowmelt for all sites except AB. Figure 13 indicates that, when considering conditions other than fresh or icy snow, PC density generally increases with increasing micro-topography except for AC where the overstory resulted in positive biases when estimating of micro-topography.



**Figure 11.** Validation of average snow depth change over transects for successive (~weekly) measurements.





**Figure 12.** Validation of average snow depth over transects for successive (~weekly) measurements.





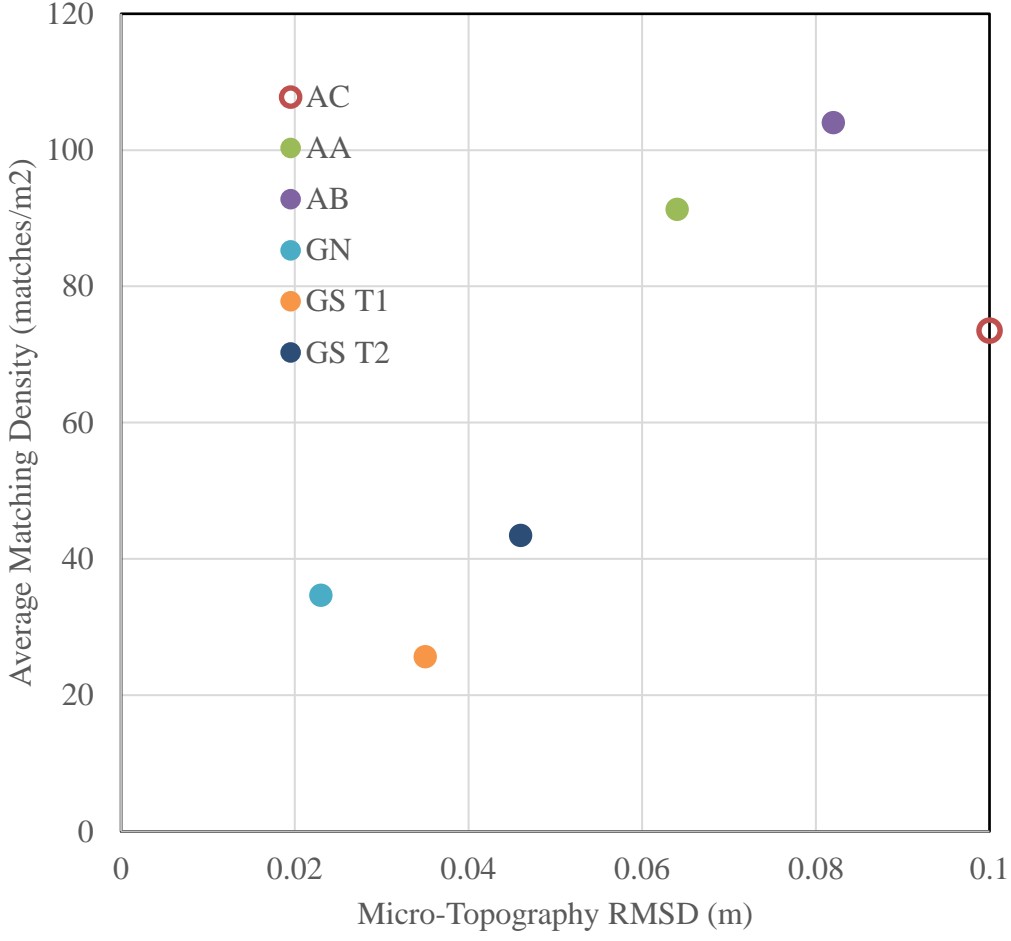

**Figure 13. Season average key point matching density (excluding fresh and icy snow) versus RMSD elevation deviation along transect. AC had an RMSD of 0.42m but is depicted at RMSD 0.1m for display purposes.**



## 4.0 Discussion

### 4.1 Temporal and Spatial Variation of Snowpack Conditions

Missions were conducted over a range of snowpack conditions, including peak snowpack, with both fresh and aged snow, ice covered snow, partial snow cover during melt and snow free just after melt.  In this sense, the experiment offers a realistic sampling of ephemeral snowpacks for the temperate climate regions of our study sites.  In contrast to studies reviewed in Section 1, snow pack conditions were often icy (5 of 29 dates) and patchy (6 of 29 dates) due to frequent rain on snow events.

The uncertainty of in-situ $\Delta SD$ was primarily due to precision error from spatial variability rather than measurement error. This aspect is important when evaluating image based estimates of $\Delta SD$ since the difference between a single in-situ and remote measurement will include some element of spatial uncertainty due to differences in the compared area.  A number of previous studies have directly reported the RMSD between image based $\Delta SD$ and point measurements (e.g. Nolan et al., 2014; Harder et al., 2016; Vander Jagt et al., 2016).  One may argue that single measurement comparisons includes the horizontal

uncertainty of the image based map but practically speaking users of a map of $\Delta SD$ are interested in the transect average in the same manner that users of current in-situ networks require transect averages rather than the spatial distribution of $\Delta SD$ at cm resolution.  Nevertheless, the within transect range of $\Delta SD$ from both in-situ and image based approaches is important for understanding the representativeness of the measurements as well as potential biases.  In this regard, the within transect variation for image based $\Delta SD$ was approximately the same magnitude as for in-situ $\Delta SD$ but skewed towards lower $\Delta SD$

when considering snow depth due to local positive biases in the snow free DSM in the presence of vegetation.  Similar biases have been reported in previous studies (Vander Jagt et al., 2016; Gindraux et al., 2017).

### 4.2 SfM Performance due to Snowpack Condition and Micro-topography

The mission performance of the consumer-grade UAV was encouraging given that it was often operated at the edge of its performance envelope in terms of wind speed and air temperature and under varying illumination conditions.  The percentage of calibrated images and matching density decreased substantially (by a factor of ~10) in the presence of precipitation or very smooth surface conditions such as fresh snow or ice.  Bühler et al. (2017) and Gindraux et al. (2017) reported similar findings with other UAV systems for fresh snow but not for ice.  However, in our study ice was typically in the form of a flat surface

pond or smoothed snow pack while in Gindraux et al. (2017) ice was the surface of a glacier that included topographic roughness. In any case, the lower matching density in both their study and ours was due to smooth surfaces.  In principle one




could interpolate $\Delta SD$ across smooth regions using the $\Delta SD$ at their perimeter but micro-topography would result in errors. Otherwise, the percentage of calibrated images did not vary substantially across sites and was consistently not a limiting factor in terms of performance (i.e. >97%).

Key point matching density ($D$) decreased by almost one order of magnitude when comparing missions flown with snow more than 1 day old and missions with either deep fresh snow or smooth icy snow packs. Previous studies have identified the drop in both elevation and SD accuracy due to deep fresh snow (Nolan et al., 2014; Avanzi et al., 2017) and icy conditions (Gindraux et al., 2017). Here we demonstrate that $D$ may be a useful indicator of such conditions and hence an indicator of the quality

of $\Delta SD$ estimates. The experiment did not control for sky conditions. The one pair of missions with similar snow conditions but different sky conditions did not show substantial changes in either the percentage of calibrated images or $D$. Nevertheless, the lack of dense canopy conditions and controlled sky conditions means that this study does not address the issue of large cast shadows (or lack thereof) on estimating snow depth changes using a low flying UAV. Bühler et al. (2017) reported that digital surface models from UAV images acquired in cast shadows appeared to be qualitatively noisier than those without shadows

and resulted in unrealistic (both negative and very high) estimates of SD after differencing from accuracy bare earth models. They suggested that a combination of visible and near-infrared imagery might reduce uncertainty in areas of cast shadow. Alternatively, measurements during overcast conditions may be sufficient to map $\Delta SD$ with sufficient accuracy in areas of persistent shadows.

Previous studies have not systematically evaluated the sensitivity of $\Delta SD$ estimation to micro-topography or vegetation density. The sites selected for this experiment were nominally flat at length scales of tens of metres, except in the vicinity of GS T3. However, micro-topography varied between sites. All of the Gatineau sites had little or no micro-topographic variation while the Acadia sites progressed from tree stumps (AA) to mounds covered with shrubs (AB) to mounds covered with shrubs and a regenerating canopy (AC). Overstory vegetation cover was less than 10% along transects except at AC where cover

within a 1 m radius vertical cylinder centred at each stake was estimated to average 38% ([0%, 52%]). However, GN and GS T1 has substantial thatch under the snow that was present during the snow free mission while AC had cover of understory herbs low shrubs ranging from 5cm to 10cm in height. As such, this experiment provides new results for a range of micro-topography and understory/low vegetation but is limited in terms of over story cover. As previously indicated, this was a conscious decision due to the difficulty of adequate non-destructive in-situ sampling in forested areas and our desire not to

further complicate the point cloud processing when having to deal with snow on vegetation. Excluding fresh and icy snow, that varied in frequency between Gatineau and Acadia, $D$ was generally proportional to micro-topographic roughness for sites without overstory. The fact that $D$ overstory (AC) may have been due more to our inclusion of vegetation PC points within our micro-topography index since the matching density at AC was similar to AB where the understory and surface topography was subjectively similar. Assuming this is the case, these results suggest a compensating effect between increasing variability



in $\Delta SD$ due to micro-topographic complexity and increasing $D$ that may explain why, outside of icy and fresh snow, RMSD and accuracy was similar across sites when estimating $\Delta SD$ change.

The geolocation performance of derived DSMs was exceptional considering that the UAV was a consumer grade device. Bias errors were smaller than the precision of the GCPs themselves suggesting that spatial variation in DSM errors may have a large random component. We could not test this hypothesis as we had limited control points that were all in relatively open areas. The DSM accuracy over GCPs was higher than reported in other studies over natural landscapes (e.g. Nolan, 2015; Harder, 2016; Gindraux 2017) but similar to performance over man-made targets (Nasrullah et al., 2016). This is partly explained by
the high spatial resolution of the imagery in our study but we hypothesize it was also due to use of easily visible elevated GCP targets that were identified in many images. For example, the number of image matches at GCPs ranged from 10 to 30. This corresponds to a theoretical ratio of between 0.9 to 5 between vertical and horizontal accuracy at a single point or 0.42 to 2.2 over five GCPs assuming independent errors at each GCP while the observed ratio based on a robust line fit was 1.1. The strong correlation in horizontal and vertical accuracy error was expected given the theoretical error model. We did not have
sufficient spatial sampling of surface elevations over snow covered areas to test the model in terms of snow surface elevation. This should be performed in future studies using reference measurements from surface instruments (e.g. Avanzi et al., 2017).

    Validation of weekly $\Delta SD$ indicates that bias across all sites and dates was smaller than the typical uncertainty for a given transect both from in-situ or image based methods and of the same order of magnitude of conventional automated or manual
measurements at point locations. There was evidence of two larger (>5 cm) over and underestimates at the forested AC site that may be due to snow present on vegetation near the ground (overestimates) or under sampling of the PC due to fresh snow (underestimates). There were also instances of underestimates exceeding 5 cm during melt over the Gatineau sites. Both of these cases corresponded to icy anterior conditions that may have favoured point cloud matches in areas with rougher snow that had not yet melted. In each of these cases, one of the compared elevation surfaces had far lower $D$ that typical for the site
suggesting that $D$ may be a useful indicator of confidence in the estimated $\Delta SD$. Notwithstanding these issues, the typical uncertainty of $\Delta SD$ was close to the theoretical error of ~2.44 cm for a single estimate. This suggests that sources of error within a transect are likely correlated since one would expect substantial reduction in the $\Delta SD$ for the transect considering that 100s of PC samples are averaged. The correlation is potentially explained by the fact that the stakes in each transect share the same images for the most part and therefore potentially suffer the same lateral displacement errors.

    Validation of $SD$ (comparing snow and snow free conditions) indicated that the range of RMSD (from ~1.5cm to 10.5cm) and bias (from 5.5cm to -10.05cm) is in accordance with the +/-10 cm uncertainty reported in previous studies (see Section 1); with a tendency for underestimation in areas with substantial ground thatch layer. The underestimation in these conditions was approximately the same magnitude of the thatch height leading us to hypothesize that they are related to an overestimate

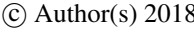



in the local DSM height as previously suggested (Nolan et al., 2014; Avanzi et al., 2017). This hypothesis could be directly validated using supplementary in-situ elevation measurements (e.g. Avanzi et al., 2017) although it is also consistent with the relatively unbiased estimate of $\Delta SD$ changes at these sites. We also hypothesize that the overestimate at AC may be due to snow covered vegetation being included in the sampled PC around each stake when estimating the DSM for snow covered

5    areas. Harder et al. (2016) noted a similar bias due to stubble protruding from shallow snow packs. Here, we used the median snow surface elevation based on PC colour processing that seemed to avoid this effect for other sites. More sophisticated algorithms for separating snow covered surfaces from over story vegetation should be evaluated.

## 5.0 Conclusions

Snow depth is an important geophysical quantity that exhibits substantial variation in space over distances of metres and in time over daily intervals. Systematic snow depth monitoring to date has emphasized temporal resolution. This study evaluated the potential for light-weight UAV imagery, processed using off-the-shelf SfM software, for mapping the change in snow depth over natural vegetated landscapes. The primary goal of this study was to compare this approach when mapping changes

15    in snow depth between snow covered dates versus between a snow covered and snow free date over land cover with varying vegetation density and micro-topography and with ephemeral snow packs. The sampled sites exhibited only modest variation in overstory vegetation cover (from 0% to 38% averaged over a transect) but substantial variability in micro-topography including tree stumps, hummocky terrain and mowed pasture. The study also addressed a second goal of comparing observed accuracy and precision with estimates based on photogrammetric theory.

A total of 71 UAV missions were flown in a range of conditions with surface elevation maps derived at between 2 cm and 3 cm horizontal ground sampling distance and with median (range) of horizontal and vertical uncertainty of 1.87 cm (0.44 cm to 11 cm) and 1.02 cm (0.045 cm to 4.6 cm) respectively in comparison to man-made ground control points. Validation over five different study sites from mid-winter to snow free conditions indicated an uncertainty of 6.45 cm (1.58 cm to 10.56 cm)

25    and accuracy of 3.33 cm (-10.05 to 5.05 cm) for the average snow depth over a ~50m long transect. Snow depth was systematically underestimated over sites with dense low vegetation by ~5 cm. As the underestimate was the same magnitude as the vegetation height during snow free conditions we hypothesize the underestimate is related to an overestimate of the snow free ground elevation. Validation for the average change of snow depth over a transect between successive (~weekly) missions indicated uncertainty of 3.40 cm (2.54 cm to 8.68 cm) and accuracy of 0.31 cm (-0.19 cm to 0.80 cm).

Observed uncertainty for weekly snow depth change agreed with the theoretical uncertainty (mean value of 2.44 cm and range of 1.2 cm to 5.25 cm depending on the number of matches at a key point). The largest uncertainties were related to comparisons




where one or both dates had low point cloud densities. This suggests that point cloud density may be a useful indicator of precision of snow depth change estimates. The observed uncertainty in snow depth was larger than that for snow depth change chiefly due to bias in estimates of the bare ground elevation in the presence of vegetation within the snow free reference image. In this case the bias is likely to be specific to local conditions and it may be possible to use in-situ measurements to

calibrating for this bias if UAV based estimates of snow depth are combined with in-situ measurements. Even so, the uncertainty of UAV based weekly snow depth change is comparable to typical in-situ measurements approaches suggesting that a combination of both measurements should be considered for producing high spatiotemporal resolution maps of snow depth change in complex terrain. We recommend that future studies consider the potential of using UAV information on snow depth change rather than absolute snow depth.

Further studies are required to investigate the performance of snow depth change mapping using similar UAV data in terms of sensitivity to changes in key point sampling density due to changing illumination, in terms of the precision of snow depth change estimates under denser canopies where the non-vegetated surface is substantially obscured, and to quantify performance as a function of UAV mission and SfM software parameters. Nevertheless, the results from our multi-site/multi-operator study

suggest that UAV based estimates of snow depth and snow depth change over areas corresponding to a typical in-situ transect have comparable uncertainty to current manual in-situ estimates while offering substantially greater coverage. Moreover, the technology can be applied with off the shelf equipment and software with minimal certification. While our study had a ~10ha limit due to using a single mission, spatial coverage can be extended to line of site using multiple missions or multiple cameras on the same UAV or even past line of sight given adequate certification. Moreover in-situ GPS targets may not be required if

baseline networks can be processed using post processed kinematic methods. Assuming these results are representative of wider landscapes and snow conditions we recommend that subsequent studies address the problem of combining airborne UAV survey based information on snow depth change with high temporal sampling satellite and in-situ information to improve snowpack characterization and reduce uncertainty in estimates of streamflow.

**Author Contribution**

RF designed the experiment, performed observations, analysis, and prepared the manuscript. CP, FC and SL designed the experiment and performed observations. MM and SO performed observations and analysis. , KH and AK performed analysis.



**Competing Interests**

The authors declare that they have no conflict of interest.

**Acknowledgements**

The authors acknowledge funding from Public Safety Canada and Natural Resources Canada and field data acquisition by Dr. Brigitte Leblon, Dr. Armand Laroque and Ms. Melanie Poirier.  The authors thank Dr. Najib Djamai, Dr. Robert Fraser, Ms. Morgan Macfarlane-Winchester for reviewing the manuscript.

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
