# Peer review of "Monitoring snow depth change across a range of landscapes with ephemeral snow packs using Structure from Motion applied to lightweight unmanned aerial vehicle videos"

_The Cryosphere, 2018_

## Referee Comment (RC1) · Anonymous Referee #1 · 23 Jun 2018

This work presents an interesting dataset of almost sixty UAV acquisitions to estimate the error under different terrain and snow conditions. Despite there are several works already published on this topic, the authors address interesting questions not very analyzed so far and the work deserves publication once some questions will be considered in a revised version.

In my opinion one of the most interesting things of the paper is to use the high number of acquisitions to relate the impact of snow conditions (mainly snow fresh and icy conditions) on the density of point clouds and the error on SD estimation. However,

[Figure]

I think that this effect should be presented in a more quantitative way than is done in the manuscript. The information presented in Figure 7 could be used for more detailed analyses and to provide mean an dispersion values for fresh snow, icy conditions and "other days". Perhaps this could be presented in box-plots being complemented with a statistical test to confiirm whether the error under the three different conditions belong to a same population. In addition to the density of points I would present the same for the error in snow depth estimation. I would also consider to compare obtained errors with wind speeds during the misions, as far as I know, this has not been addressed yet in literature in detail and your dataset is nice for this purpose.

Why does Figure 13 exclude snow fresh and icy conditions? I think they should be also included or at least to evaluate what happens when they are also included.

Other point that can be considered is that most of the plots referred to errors are based on the snow depth differences between consequtive days, and the discussion mentions that future research should focus more on this than in total snow depth. I do not fully agree in this statement, since differences between days are interesting for many purposes, but most of the snow research in areas such hydrology or glaciology (for mass balance) is more interested to have a total estimation of snow depth rather than analysing in detail changes along a now season. Thus, I would give same weight in the results to show errors for snow differences and total snow depth estimation.

In general, the paper is written very clearly but in my opinion the figure captions are very difficult to be understood by themselves, I would consider to have a look them and add the text necessary to facilitaty its understanding.

When the equipment is described, there is indicated the resolution of the camera but not about the size of the sensor and their distortion parameters, I would mention about this as probably it is more important than the resolution itself. Was the distortion of the images corrected? I am not sure if this is related with points 4 and 6 of point 18. I do not understand exactly what is meant to say in those points... Hoping my comments

will result useful

---

## Referee Comment (RC2) · Anonymous Referee #2 · 25 Jun 2018

Review of "Monitoring snow depth change across a range of landscapes with ephemeral snow packs using Structure from Motion applied to lightweight unmanned aerial vehicle videos" by Richard Fernandes et al.

This manuscript uses unmanned aerial vehicles for monitoring snow depth change. Emphasis is placed on the accuracy assessment of the snow depth change between UAV flights and on the expression of expected accuracy from conventional photogrammetric theory. Generally this manuscript provides information on the application of UAV's to quantify snow. There have been several papers in recent years which have

presented very similar types of work (which are already cited in this manuscript) so at this point for a paper to be accepted there need to be clear novel contributions. In general, specific portions of this manuscript (theoretical error estimation and snow depth change error analysis) are novel contributions but many portions are not and portions of introduction, methodology and discussion are tangential. I would recommend that this manuscript undergo major revision before being considered again. There are various spelling and grammatical errors throughout -I am not noting them here as more major changes are needed before I'd recommend acceptance. Main comments follow.

Length and Level of detail: This manuscript is very long and very detailed. Obviously enough detail is needed to allow for reproducibility but this is too much information as one must avoid having a reader lose interest if it takes too long to get to any results. In addition, the large amount of detail regarding the accuracy assessment and operations is unnecessary in my opinion as previous work on this topic have established that baseline knowledge. My suggestion would be to focus on the novel portions of the manuscript. Specifically, the work on the modelling of the expected accuracy wrt to flight characteristics as determined from conventional photogrammetric theory. This would be very helpful to in flight planning.

In situ snow depth estimation: The approach taken to estimate snow depth in the field - using snow stakes protruding from the surface and determining SD height from photos taken $\sim$ 5 m away- is problematic. SfM is based on features that can be clearly identified in multiple images. Thus a stake in a snow field will render a better sfm solution/point cloud near the stake than further away. The authors need to somehow demonstrate that their in situ SD observational protocol does not bias the point cloud accuracy or density of the sfm solution. Even if stake points are removed from the point cloud the immediately adjacent snow points will also be biased to the more precise stake solution. Are the results of snow depth change valid away from these snow stakes or not? Only if this can be successfully argued can hypothesis 1 be tested.

Specific comments:

[Figure]

Abstract: please synthesize the conclusions.

Page 2 Line 11-24: This is emblematic of the level of detail concerns I have. Is it necessary to have an explanation of the WMO SD network when the focus on the paper is SD from UAV's?

Page 3 Line 6-7: If this is not a focus of the paper why mention this?

Page 3 Line 27-28: Many examples in the literature do this already.

Page 4 Line 6-7: This has been discussed and an example is given in: Schirmer, M. and Pomeroy, J. W.: Factors influencing spring and summer areal snow ablation and snowcover depletion in alpine terrain: detailed measurements from the Canadian Rockies, Hydrol. Earth Syst. Sci. Discuss., https://doi.org/10.5194/hess-2018-254, in review, 2018.

Page 5 Line 18-19: Why?

Page 5 Line 28: Are these actually 5 study sites? I was expecting 5 sites with different features and locations based on all the preceding text. This seems like 2 sites with a total of 5 stratified sample areas of analysis.

Table 1 and 2: combine

Figure 1 and 2: Google earth citation? Google earth screenshot is not typically publication quality and a better map should be provided prior to any publication.

Page 9 Line 21-26: What is the influence of GCP's being located above the surface of interest. Typically GCP's should be located at same height of surface. How were GCP locations measured? dGPS? What is the accuracy of this measurement?

Table 3: Is this table necessary as DJI Phantom Pro's are extremely popular (not an obscure UAV)?

Page 13 Line 8-9: imagery was nadir?

Page 14 Line 17-19: Clarify what was optimized.

Page 18 Line 7-8: What are the implications of this? I would expect that this would add a smoothing artefact.

Page 19: Line 6-12: These sentences are repetitive. Remove one and merge paragraphs?

Table 6. A wind speed of 26 ms-1 is crazy high to fly a UAV safely. Are these correct units? Please explain if/how this wind speed observations are different from actual flight conditions.

Figure 7: y label axe units need to be improved. Xlabels could remove the year from each date. Plot areas are also not consistent. Formatting is not publication ready.

Figure 8 and 9: combine into a) and b)? what is the meaning of circle size? Add legend.

Figure 13: Putting AC RMSD at 0.1 when it is actually at 0.42 is misleading even if noted.

Page 34 Line 17: "minimal certification" this is not mentioned elsewhere.

Page 34: Line 19-20: This was tested in Harder et al. (2016) and was determined that even with RTK corrected photo geotags GCPs were still needed.

---

## Author Comment (AC2) · 20 Aug 2018

Please refer to my Authors Comments where responses to both referees are consolidated.

---

## Author Comment (AC3) · 20 Aug 2018

Please refer to my Author's comments where my responses to both referees are consolidated.

---

## Author Response (AR1)

Authors Response to Reviewers Comments

We appreciate the comments from both reviewers. They were both insightful in terms of interpreting our data and helpful with respect to improving the presentation of the manuscript. Below is a list of itemized comments (in italics) and responses. We have also provided additional supplementary material and a separate PDF document with all figures.

Reviewer #1

1. *There are various spelling and grammatical errors throughout*

Sincere apologies. The errors are the responsibility of the first author. The revised manuscript has been checked by internal reviewers for errors. A MS word version of the manuscript with changes is available upon request.

2. *the large amount of detail regarding the accuracy assessment and operations is unnecessary in my opinion as previous work on this topic have established that baseline knowledge*

With respect to operations we have moved Table 1 and Section 2.5 to supplementary material. With respect to accuracy assessment we prefer to retain most of the detail since we feel that it is critical that others can replicate the metrics we have used in our study. Moreover, previous studies (page 27 lines 14-16) have reported metrics that may have not been relevant for our research question (e.g. the compared individual SD measurements rather than transect averages).

3. *The authors need to somehow demonstrate that their in situ SD observational protocol does not bias the point cloud accuracy or density of the SfM solution. Even if stake points are removed from the point cloud the immediately adjacent snow points will also be biased to the more precise stake solution. Are the results of snow depth change valid away from these snow stakes or not?*

This is an important question. We examined maps of Automated Keypoints produced by PIX4D Mapper for each mission. Our response is now given on page 30 lines 20-31 and provided here for convenience:

Validation of $\Delta SD$ requires minimally invasive reference estimates using methods that also does not substantially change the performance of UAV estimates. Considering the potential for large variations in $SD$ and $\Delta SD$ with microtopography we decided to control the reference locations by using fixed stakes. This strategy could have led to an (artificial) increase in precision if the stakes led to an increase in the $D$ as well as an increase if accuracy if the same keypoints on stakes were detected in multiple images within or between missions. Examination of maps of automated keypoints *a posteriori* indicated that the 10 PIX4D algorithm rarely found a keypoint along a stake (e.g. Supplementary Material Figures S1 to S5). Furthermore, the few cases where a keypoint was identified on a stake corresponded to locations with exposed vegetation around the stake that would potentially exhibit a match in any event. PIX4D Mapper uses a proprietary implementation of a reduced set of features derived from the Scale Invariant Feature Transformation (SIFT) (Strecha, 2011). SIFT features are defined to specifically eliminate keypoints that have poorly determined locations but high edge responses; especially corner features (Lowe, 2004). 15 We hypothesize that, especially for snow covered conditions, the relatively narrow correspond to such features and are subsequently avoided by PIX4D Mapper when identifying keypoints. If so, our results may actually be somewhat pessimistic since there are potentially fewer keypoints in the vicinity of stakes.

4. *Page 2 Line 11-24: This is emblematic of the level of detail concerns I have. Is it necessary to have an explanation of the WMO SD network when the focus on the paper is SD from UAV's?*

The intent was the need to survey current approaches for systematic survey of SD. Specifically, with respect to their spatial coverage and uncertainty. This section has now been shortened to focus on that point (page 2 lines 1-24). Moreover, the discussion of the rationale for using 5 GCPs is now less detailed (page 8 lines 6-10).

5. *Page 3 Line 6-7: If this is not a focus of the paper why mention this?*

These lines have been removed.

6. *Page 3 Line 27-28: Many examples in the literature do this already.*

These lines are a paraphrase of the recommendation of the study of de Michele et al. 2016; a rather recent study. We agree that since 2016 there is a growing body of studies that are evaluating the user of UAV for SD mapping using, for example SfM. Page 3 lines 1-29 survey the literature. We do not feel there is anything wrong in performing additional validation studies for methods as long as they serve as replicate attempts of previous studies that have not been sufficiently replicated and/or include novel elements. The novel elements in our study are discussed on Pages 3 and 4 in terms of issues we aimed to address.

7. *Page 4 Line 6-7: This has been discussed and an example is given in: Schirmer, M. and Pomeroy, J. W.: Factors influencing spring and summer areal snow ablation and snowcover depletion in alpine terrain: detailed measurements from the Canadian Rockies, Hydrol. Earth Syst. Sci. Discuss., https://doi.org/10.5194/hess-2018-254, in review, 2018.*

We have added this reference on page 3 line 33.

8. *Page 5 Line 18-19: Why?*

We assume the question "why" refers to our phrase "except for very smooth snow pack conditions". We have added add explanation for this exception on page 5 lines 11-13.

9. *Page 5 Line 28: Are these actually 5 study sites? I was expecting 5 sites with different features and locations based on all the preceding text. This seems like 2 sites with a total of 5 stratified sample areas of analysis.*

As this is a question of nomenclature we have not made changes but confirm that our use of the term "site" in consistent. We use the term "site" since each area had a distinct land surface condition (cover and/or topography) and were surveyed using separate UAV flights. The fact that they are proximal takes nothing away from differences in microtopography and vegetation cover. In contrast we would have used a term "sample area" if we were considering replicates within the same surface and climate conditions. As a further note, we use the term "study regions" to imply areas separated sufficiently to expect different climate conditions.

10. *Table 1 and 2: combine*

Tables combined.

11. *Figure 1 and 2: Google earth citation? Google earth screenshot is not typically publication quality and a better map should be provided prior to any publication.*

Map and citation of map improved. We have combined Figures 1 and 2.

12. *Page 9 Line 21-26: What is the influence of GCP's being located above the surface of interest. Typically GCP's should be located at same height of surface. How were GCP locations measured? dGPS? What is the accuracy of this measurement?*

We did not test the influence of variation of GCP height so we cannot answer the question. Our rational for GCPs above the surface was to avoid artificially increasing the accuracy of SD estimates by identifying locations on the snow surface as control points. We mention this now on page 8 lines 20-21. The total uncertainty of the GCPs are given on page 8 lines 15-18. The accuracy is less than this amount but is not included since the total uncertainty is very close to the GSD of our data. Measurement of GCPs is given in detail in Prevost 2016a,b. As the focus of the paper is not on GCP measurement we have only added mention of the generic approach for GCP processing and the equipment used on page 8 lines 17-18.

13. *Table 3: Is this table necessary as DJI Phantom Pro's are extremely popular (not an obscure UAV)?*

It is convenient for reference since the instrument in theory can be modified. The table has been placed in Supplementary Material and referred to on page 11 line 7.

14. *Page 13 Line 8-9: imagery was nadir?*

Yes. We have noted this now on page 11 line 19.

15. *Page 14 Line 17-19: Clarify what was optimized.*

We used the wrong term. We should have used the term "model" since the mission parameters were varied to model the sensitivity of height uncertainty to parameter combinations. The change is made on page 12 line 27.

16. *Page 18 Line 7-8: What are the implications of this? I would expect that this would add a smoothing artefact.*

Removing all points corresponding to dead and live vegetation above the soil would result in smoothing of the index of micro-topography we used. It is for this reason that we only removed points more than the height of the observed maximum transect average snow surface elevation. By doing so we preserve roughness elements that are covered by snow at some point during the season (i.e. that contribute to topographic effects). This is noted on page 15 lines 12-13. Our index of micro-topography was intended to be easy to understand and replicate and relatively robust to between site differences in the summer UAV based elevation model used.

*17. Page 19: Line 6-12: These sentences are repetitive. Remove one and merge paragraphs?*

The sentences are similar but distinct in that lines 7-9 refer to the median of points identified as snow covered according to the criteria on lines 3-4 whiles lines 10-12 refer to the median of all points since in this case we are interested in snow free ground. We have not made any change.

*18. Table 6. A wind speed of 26 ms-1 is crazy high to fly a UAV safely. Are these correct units? Please explain if/how this wind speed observations are different from actual flight conditions.*

Wind speed units were wrong. They should have been km/hr. We have changed the text and the table headers. We also found that we cited the maximum UAV speed in the text but not the speed it was operated at (3.5m/s). We have clarified this in Table 2.

*19. Figure 7: y label axe units need to be improved. Xlabels could remove the year from each date. Plot areas are also not consistent. Formatting is not publication ready.*

Labels, areas and formatting improved.

*20. Figure 8 and 9: combine into a) and b)? what is the meaning of circle size? Add legend.*

Combined into new Figure 7. Circle area is proportional to key point match density. The caption has been updated.

*21. Figure 13: Putting AC RMSD at 0.1 when it is actually at 0.42 is misleading even if noted.*

We have modified the figure to include an x-axis break. The figure now corresponds to Figure 9.

*22. Page 34 Line 17: "minimal certification" this is not mentioned elsewhere.*

We have removed this phrase since it is not directly relevant to the research goals or issues identified in the introduction. Moreover, certification requirements so to be changing over time and with jurisdiction.

23. *The information presented in Figure 7 could be used for more detailed analyses and to provide mean an dispersion values for fresh snow, icy conditions and "other days". Perhaps this could be presented in box-plots being complemented with a statistical test to confiirm whether the error under the three different conditions belong to a same population. In addition to the density of points I would present the same for the error in snow depth estimation. '*

We agree that the accuracy as a function of snow condition is important from both a practical perspective and to explain the cause of low key point matching density conditions. We have modified Figure 7 (new Figure 9) by including icy/fresh snow data; we also removed snow free data from the "other" cases since our goal is to look at key point matching density for conditions with snow. However, we did not include box plots since the sample sizes for icy/fresh snow are both small and differ between regions (6 for Gatineau and 3 for Acadia). Instead, we performed a test for difference of means in point cloud density between icy/fresh snow and "other days" for each site. This test implicitly accounts for sample size and the effect of snow condition considering measurement error and natural variation. We report on this test on page 20 lines 25-26 and in the Figure 9 caption. We also performed the same test for snow depth but found no significant differences due perhaps to a decrease in snow depth variation during icy and fresh snow. Since statistics are not as relevant due to sample size issues, we discuss individual effects on page 30 lines 22-28.

24. *I would also consider to compare obtained errors with wind speeds during the misions, as far as I know, this has not been addressed yet in literature in detail and your dataset is nice for this purpose.*

A priori we did not consider this factor so we did not perform replicate trials with wind speed changing and other conditions (especially illumination and snow condition). Following the suggestion of the reviewer we evaluated regressions of key point density, geolocation performance and snow depth change performance as a function of wind speed (maximum wind speed for snow depth change). Our findings are briefly discussed on page 24 lines 8-11. We mention this limitation of our experiment in terms of wind speed replicate trials on page 29 lines 10-20 as well as the fact that it may not be important given that PIX4D seems to provide very similar results with or without UAV ephemeris.

*25. Why does Figure 13 exclude snow fresh and icy conditions? I think they should be also included or at least to evaluate what happens when they are also included.*

We have now included these conditions (new Figure 9).

*26. I would give same weight in the results to show errors for snow differences and total snow depth estimation.*

We have provided the same detail and annotations in results (new Figure 10) for both quantities and the same statistics (page 25 lines 11-14 and lines 22-26). We acknowledge that our discussion is more detailed for snow depth differences than snow depth estimation. The issues is that, other than a systematic bias noted for snow depth estimation, we did not notice any specific pattern in its errors as a function of snow condition or even as a function of site. Since our results for snow depth estimation were similar to other studies we left our discussion at that rather than trying to tease out patterns that for which we did not have statistical or physical explanations.

*27. the figure captions are very difficult to be understood by themselves, I would consider to have a look them and add the text necessary to facilitaty its understanding.*

We have added text to most figure captions.

*28. When the equipment is described, there is indicated the resolution of the camera but not about the size of the sensor and their distortion parameters, I would mention about this as probably it is more important than the resolution itself. Was the distortion of the images corrected?*

PIX4D processing includes accounting for camera distortion parameters both as initial conditions for bundle adjustment and as a refinement during bundle adjustment. As Reviewer#1 requested we reduce generic details regarding operations we hesitate to include discussion of how camera distortion is handled in the main body of the manuscript. Rather, we have discussed this in Supplementary material Section 2.

[revised manuscript text omitted]
 digital images from an aircraft to map SD over bare surfaces at between 6 cm and 20 cm horizontal resolution and 10 cm vertical root mean square difference (RMSD) compared to in situ estimates. Similar results were reported using airborne systems over prairies (Harder et al. 2016), alpine shrub lands (Vander Jagt et al., 2015; De Michele et al. 2016; Harder et al. 2016) and glaciers (Gindraux et. al., 2017). Photogrammetric theory suggests that, with all other factors constant, both horizontal and vertical uncertainty in digital surface models is proportional to imaging altitude above ground level (a.g.l.) and inversely proportional to image overlap (Forstner, 1998). Aircraft systems offer wide coverage but are often unable to fly low and slow enough to result in either horizontal or vertical uncertainty of SD comparable to in situ methods. For example, Gindraux et al. (2017) indicated that their final digital surface model (DSM) horizontal resolution of 50 cm based on 6 cm ground sampling distance (GSD) airborne imagery resulted in inconclusive results between observed surface roughness and DSM uncertainty. Bühler, et al. (2016) used an octocopter UAV with a high performance geolocation system flying between 97 m to 113 m a.g.l. to map SD at ~10 cm GSD. They reported a RMSD between 7 cm and 15 cm for flat alpine meadow and exposed rock and up to 30 cm for tall grass. Avanzi et al. (2016) used SfM applied to ~2 cm GSD quadcopter imagery over glaciers to map surface elevation with a root mean square difference (RMSD) under 5 cm averaged over 135 manual measurements within a 1ha region. If some of the uncertainties in UAV are persistent over time then differences in $\Delta SD$ between UAV and in situ measurements may be even less than the 5 cm reported by Avanzi et al. (2016). Photogrammetric theory also indicates that vertical uncertainty from SfM should be proportional to the density of matching keypoints between overlapping images (Forstner, 1998). Keypoint density will~~

~~change with snowpack and surface conditions. Gindraux et al. (2017) reported a two order of magnitude decrease in keypoint density for a fresh snow covered glacier in comparison to one day old snow cover. Current studies have considered only a limited range of snowpack conditions (usually cold snowpacks prior to melt), micro-topography and vegetation cover (usually areas without any vegetation above the snowpack).~~

The issues that remain to be addressed regarding UAV based mapping of SD require multiple experimental treatments including climate and snow conditions that cannot easily be controlled and land surface conditions that can be controlled. Here we chose to control the survey methodology by using a single low-cost commercially available solution for UAV based mapping of three dimensional point clouds and select mission parameters that should maximize the accuracy of elevation estimation based on photogrammetric theory, even if the solution may not be optimal in the sense of logistical constraints of time or cost. Secondly we select sites with a range of micro-topography and vegetation cover but limit vegetation cover to ≤50% and only validate SD in openings. This strategy simplifies the approach used to extract surface locations within three dimensional point clouds leaving the issue of UAV based SD mapping under closed canopies for further study. Thirdly we locate the sites within regions of ephemeral snowpack since this should correspond to a worst case assessment of uncertainty, especially with respect  to $\Delta SD$. Given these limitations, the initial broad research question regarding snow depth mapping is refined into two specific research questions addressed in this study:

~~This study evaluates a baseline solution for mapping $\Delta SD$ using one widely used commercial UAV imaging system (the Phantom 3 Professional, P3P https://www.dji.com/phantom-3-pro/info) and one implementation of SfM software (PIX4d Mapper, https://pix4d.com/product/pix4dmapper-photogrammetry-software/ ). Even with this limited scope there are a number of potential experimental treatments including parameters for UAV missions, software parameters, micro-topography and vegetation cover. Here we control the first two treatments by selecting 
[revised manuscript text omitted]
 vegetation aalso acts to bias estimates of $S\Delta SD$ (Harding et al. 2016)from our UAV approach. Mmicro-topography was quantified as the deviation from a local robust linear slope trend (MATLAB function 'lmfit' with robust option, https://www.mathworks.com/help/stats/fitlm.html) with a 15 m moving window oriented along the transect. Deviations greater than the maximum snowpack elevation at each transect during the season were removed when computing the root mean square deviation RMSD over a transect to eliminate overstory vegetation that normally would be above the snowpack.

**2.7 Elevation and Overstory Cover Extraction**

[revised manuscript text omitted]
 factor in terms of performance (i.e. >97%). Matching density were consistent within Acadia and ~70% higher than Gatineau for conditions without fresh snow or ice. Examination of match points suggests that the higher matching density over Acadia was due to a combination of increased microtopographic features and increased vegetation in comparison to Gatineau. This result suggests a potential compensating effect for the image-based approach whereby more matches are found in landscapes where DSM variation, and possible SD variation, is greater. However, we have not considered closed canopies where occlusion due to other canopy elements may result in a saturation of matches as vegetation density or cover increases.~~

~~The number of images matches per point (K) was not rigorously addressed since there is no means of extracting this information for a sub image within Pix4D other than by manual examination of individual images (there are ~1000 images per mission). Average values or K for each mission (4-10) fell within or above the range observed in our trial missions (4.3-7.4). The higher values of K were observed over Acadia where some vegetation and stumps were matched in almost all overlapping images. In contrast, for Gatineau the average values of K ranged from 4-5. The similarity between our initial estimates of K from four trial flights and actual values for K suggests that the theoretical prediction of accuracy should also be in general agreement with observed accuracy for $\Delta SD$.~~

[revised manuscript text omitted]
 $\Delta SD$ when comparing snow and snow free conditions indicates a linear relationship with a tendency for underestimation of in-situ estimates of $\Delta SD$ in areas with substantial ground thatch layer.  The underestimate in these conditions was approximately the same magnitude of the thatch height leading us to hypothesize that they are related to an overestimate in the local DSM height.  GCPs should be placed along the snow stake transects to test this hypothesis.   We also hypothesize that the overestimate at AC may be due to snow covered vegetation being included in the sampled PC~~

around each stake when estimating the DSM for snow covered areas. The use of a median snow surface elevation may not have been sufficient to remove bias at this site. 
[revised manuscript text omitted]

Tables

---

## Author Response (AR2)

**Responses to Revision#1 of Manuscript tc-2018-82**

**Responses to Editor's Comments**

**1. P4.15 parenthesis not closed**

Response: Agreed.

Action: Corrected.

**2. P5.8 delete space before comma.**

Response: Agreed.

Action: Corrected.

**3. Are all table necessary in the main article? Think if some (Table 3 and 4 in particular) could be moved to the supplement.**

Response: Agreed.

*Action: Moved.*

**4. P20.13. "least absolute residual regression" does not sound familiar to me. Double check.**

Response:  least absolute residual regression was used to highlight the accuracy of the SD change estimates once outliers due to icy/smooth snow were discounted.

Action: added reference to algorithm used (https://www.mathworks.com/help/stats/robustfit.html).

**5. Figure 10a. Y-axis title should be "change"**

Response: Agreed.

Action: Corrected.

**6. P27.17 "users of" (I think)**

Response: Agreed.

Action: Corrected.

**7. P33.4 isolated coma to be removed.**

Response: We did not find an isolated comma, but found an extra period after Ambio. I think the confusion is that the title of the article has an Oxford comma before the and.

Action: Corrected the extra period.

**8. That said, anonymous Referee #2 is right in suggesting that a shortened version of the manuscript would have more impact on its readers. I understand this come rather late in the review process and I do not want to condition publication to the 25% shortening suggested by reviewer #2. However, when doing a final check (some typos remain), I recommand that you try to make the text sharper.**

Response: Agreed.

Action: The following edits were applied to the abstract, introduction and methods to reduce the length of these sectiond from 21 pages to 17 pages:

Removed four sentences from the introduction between 2.13 to 3.11 regarding other approaches for in-situ snow depth monitoring (there is still a list with references of these methods.)

Shortened the rationale of our GCP target selection on 8.3 to 8.8.

Moved details on GCP targets from 8.19 to 9.5 to Supplementary Material S1.

Moved details on GCP targets from 10.4 to 10.20 to Supplementary Material S2.

Moved 11.25 to 11.27 discussing flight duration to Supplementary Material S3.

Combined sentences at 16.3 to 16.4.

Moved 18.7 to 18.10 discussing failed missions to Supplementary Material S6.

**Response to Comments from Reviewer #1**

**9. As is this paper is very long and it should be further focused on the key findings revolving around the 2 research questions and much of the methodological and supporting details moved to the supplement.**

**From the manuscript types considered by The Cryosphere it states "Research articles report substantial and original scientific results within the journal's scope. Generally, these are expected to be within 12 journal pages, have appropriate figures and/or tables, a maximum of 80 references, and an abstract of 150–250 words." There is always a difference between the submitted manuscript and type set version but I believe that this article is well in excess of these guidelines**

Response: Agreed.

Action: See response to comment #8.

**Response to Comments from Reviewer #1**

There were no comments requiring action.